# An Approach toward 17-Arylsubstituted Marginatafuran-Type Isospongian Diterpenoids via a Palladium-Catalyzed Heck–Suzuki Cascade Reaction of 16-Bromolambertianic Acid

**DOI:** 10.3390/molecules27092643

**Published:** 2022-04-20

**Authors:** Yurii V. Kharitonov, Elvira E. Shults

**Affiliations:** N.N. Vorozhtsov Novosibirsk Institute of Organic Chemistry, Siberian Branch of the Russian Academy of Sciences, Academician Lavrentyev Ave, 9, 630090 Novosibirsk, Russia; khariton@nioch.nsc.ru

**Keywords:** isospongian-type diterpenes, furanolabdanoids, lambertianic acid, palladium-catalyzed Heck–Suzuki cascade reaction, diastereoselectivity

## Abstract

Isospongian diterpenes are a small but growing family of natural tetracyclic secondary metabolites isolated from marine organisms, primarily sponges and nudibranchs. A palladium-catalyzed domino Heck–Suzuki reaction sequence for the synthesis of the tetracyclic skeleton of marginatafuran-type isospongian diterpenoids with a wide variety of substituents in the C-17 position is reported. The proposed approach was based on selective transformations of the accessible plant diterpenoid lambertianic acid and includes an intramolecular Heck reaction of 16-bromolambertianic and arylation of the palladium intermediate with arylboronic acid. The influence of the nature of the substituent both in arylboronic acids and in the furan ring of 16-bromolambertianic acid on the direction and chemoselectivity of the reaction has been studied. The described derivatization of natural furanolabdanoid lambertianic acid produced new functionalized molecules for biological study and gave novel insights into the reactivity of complex molecular structures.

## 1. Introduction

Marine sponges have been considered as a very remarkable field for the discovery of bioactive natural products. Among sponge metabolites, spongian diterpenes have received diverse attention due to their role as eco-physiological mediators. Being devoid of the physical protection, marine sponges are obvious targets for predation, and as a consequence, many spongian diterpenes have been isolated from spongivorous marine opisthobranch mollusks (nudibranchs) that predate upon these sponges. In particular, the furanoditerpenes marginatafuran **1** (Figure 1) presented the first example of an isospongian-type diterpenoid, which was isolated from the skin extract of the northwestern Pacific common dorid nudibranch *Cadlina luteomarginata* [1]. In fact, compound **1** was later found in minor amounts in sponges from the same area belonging to the genus *Aplysilla* [2]. Another furanoditerpene marginatone **2** was isolated from the sponge *Aplysilla polyrhaphis* [3], and *A*. *glacialis* [4], together with its derivative 20-acetoxymarginatone **3**. Compound **3** was also isolated from skin extracts of a marine gastropod mollusk *Cadlina* luteomarginata [5].

Owing to the rare marginatane carbon skeleton and biological profiles, these natural products represent attractive targets for the synthetic communities. The first synthetic studies use the polyene substituted furan ambliofuran [6] as the starting compound [7,8]. Using the mercury(II) reagent, ambliofuran was cyclized, leading to the tetracyclic isospongiane in 13% yield. Compound with the marginatane carbon skeleton was obtained after the demercuration treatment with sodium borohydride [7]. Ambliofuran had also been cyclized to furanoditerpene using SnCl_4_ as an electrophile initiator [8]. Indium tribromide-promoted epoxy olefin cyclization with the formation of the tetracyclic marginatane-type compound was described in [9]. An oxidative free-radical cyclization of a polyene compound with a mixture of Mn(OAc)_3_ and Cu(OAc)_2_ was used to provide stereoselectively the tricyclic intermediate, whose functional group manipulation and homologation at C13 allowed the construction of the required furan ring D of the marginatane carbon skeleton [10]. A unified synthetic route toward three isospongian diterpenoids (–)-marginatafuran **1**, (–)-marginatone **2** and (–)-20-acetoxymarginatone **3** was developed in which an intramolecular Diels–Alder cycloaddition reaction and a ring-closing metathesis reaction were used as the key operations to construct the required polycyclic skeleton [11]. Isospongian-type diterpenoid **3** was also synthesized starting from labdane-type diterpene (+)-coronarin E, using minor modifications of reported procedures, which included regioselective hydrogenation and stereocontrolled-intramolecular electrophilic cyclization [12,13]. Coronarin E has been isolated from various medicinal plants [13] and also synthesized from the available natural compound (-)sclareol [14]. Herein, as a continuation of our previous work on the synthetic transformation of labdanoid-type diterpene lambertianic acid **4** [15,16,17,18,19,20], we wish to describe the syntheses of 17-arylsubstituted marginatafuran-type isospongian diterpenoids.

The palladium-catalyzed domino (cascade) reaction of 16-bromolambertianic acid derivatives, including the intramolecular Heck reaction and cross-coupling Suzuki reaction with arylboronic acids, became the main synthetic method. The effect of the varying substitution pattern of the boronic acid aromatic ring, as well as the nature of the C-15 substituent in the terpenoid skeleton, will be explored. Previously, this Heck–Suzuki cascade reaction strategy has been intensely investigated in the synthesis of a variety of heterocycles, carbocycles [21,22,23,24], and also applied toward the synthesis of various natural products [25]. Due to the formation of a new chiral quaternary stereocenter, at the first stage as a result of the intramolecular Heck reaction, the products of this domino reaction are formed as a mixture of two diastereomers. Obtaining optically active compounds requires the use of chiral catalysts [26,27,28,29,30]. In the case of lambertianic acid, a reaction with high diastereoselectivity is expected due to the higher steric availability of the α-side of the diterpene core [31].

## 2. Results

Bromination of lambertianic acid **4** (the main component of the pine oleoresin of P. sibirica J. Mayr) [32] in a CH_2_Cl_2_ solution with NBS at room temperature afforded 16-bromolambertianic acid **5** (isolated yield 40–64%). The reaction of bromide **5** with phenylboronic acid **6а** (1.2 equiv.) in the presence of Pd(PPh_3_)_4_ (2 mol%) as catalyst and K_2_СO_3_ (3.6 equiv.) as the base in aq. DMF proceeds by heating at 80–85 °C for 24 h with the formation of 17-phenylisospongian-13(16),14-dien-18-oic acid **7а** as the main product in the isolated yield 71% (Figure 1).

In the found conditions, the reaction of bromide **5** with 2-fluorophenylboronic acid **6b** or 4-fluorophenylboronic acid **6c** led to the formation of two products: corresponding cascade reaction products **7b**,**c** and the Suzuki cross-coupling product **8b**,**c** (Table 1, entries 1,2). The reaction products were separated by column chromatography. A significant effect of the position of the substituent in the aromatic ring of boronic acid on the direction of the reaction was observed. Thus, in the reaction of **5** with 2-fluoroboronic acid **6b**, the product of arylation **8b** and domino reaction **7b** were isolated in 45% and 23% yields, respectively (entry 1). In cross-coupling of **5** with 4-fluoroboronic acid **6c**, the isospongian-type compound **7c** was isolated as the main product (40% yield). The yield of the Suzuki coupling reaction product **8c** was about 20%. When carrying out the reaction of bromide **5** with **6b** at 60–65 °C, incomplete conversion of the starting compound **5** (60–70%) was observed, while the ratio of the resulting products **7b**,**8b** remained practically unchanged. A low conversion of 16-bromolambertianic acid **5** was observed in the reaction with 3-substituted arylboronic acids (3-methoxyphenyl- and 3-fluorophenylboronic acids).

The optimization of reaction conditions was exemplified by the reaction of the bromide **5** with 4-methoxyphenylboronic acid **6d** (Table 1, entries 3–15). It was observed that no cross-coupling reaction products were obtained when reaction was carried out under Pd(PPh_3_)_4_ catalyst in the presence of cesium carbonate or tripotassium phosphate as the base. These bases along with potassium carbonate were often used in cross-coupling reactions with bromofurans [33,34]. By performing the Pd-catalyzed reaction at high temperature, the competing process of hydrodehalogenation of furanolabdanoid **5** become favored, and only lambertianic acid **4** was formed (Table 1, entry 4, conditions *a*). 

It has been known that various phosphine ligands are effective in stabilizing the Pd(0) species during the cross-coupling reaction of arylboronic acid, and an increase in the steric hinderance of the ligand in the palladium complex promotes a more rapid occurrence of the stage of incorporation into the alkene in the catalytic cycle of the Heck reaction [35,36]. It was found that sterically demanding and electron-rich monophosphine ligands –di-(1-adamantyl)-benzylphosphine ((1-Adm)_2_PBn) or 2-dicyclohexylphosphino-2’,4,’6’-triisopropylbiphenyl (XPhos) were not active in this Heck–Suzuki cascade reaction (Table 1, entries 5,6). In the reaction of terpenoid 2-bromofuran **5** with 4-methoxyphenylboronic acid **6d** under catalyst by a Pd(OAc)_2_-P(Tol)_3_ system in DMF-water, only the domino reaction product **7d** was obtained (Table 1, conditions *b*). We have shown that using bidentate ligands (chelating ligands), which can provide a predominance of reductive elimination over β-hydride shift [37], ensured an increase in the isolated yield of both compounds **7d** and **8d** (Table 1, conditions *c*–*e*). The selectivity in compound **7d** formation was increased by using sterically more demanding ligands –(*R*)-1,1′-bis(ditolylphosphino)-2,2-binapthyl (TolBINAP) (conditions *d*). The increase in the overall yield of compounds **7d** and **8d** under the catalysis by Pd(OAc)_2_-(*R*)-BINAP system (conditions *e*) was characteristic. Using a Pd(OAc)_2_-(*S*)-BINAP system led to the increasing of the yield of the tetracyclic compound **7d**; the stereoconfiguration of the formed product was the same, as in the reaction, it proceeds by using a Pd(OAc)_2_-(*R*)-BINAP system (conditions *e**). We next examined other parameters, including the study of influence of the temperature, nature of the base and solvent on the yield of the domino reaction product (conditions *f–i*). It was found that the reaction in DMF–water at a lower temperature (60–65 °C) under the Pd(OAc)_2_-(*R*)-BINAP system resulted in an increase in the selectivity of the reaction and the yield of compound **7d** (isolated yield 56%, conditions *f*). The reaction employing Cs_2_CO_3_ as the base proceeded with lower selectivity for the domino reaction product **7d** (yield 35 and 21%, conditions *g*). The further lowering of reaction temperature led to a decrease in the conversion (30%) and the selectivity and yield of reaction products **7d** and **8d** to 6% (Table 1, conditions *h*). The reaction in the СH_3_CN-H_2_О solvent system at 60–65 °C proceeds selectively but with lower yield of the domino reaction product **7d** (yield 24%, conditions *i*); additionally, at the same reaction time, the observed conversion was about 90%. Characteristically, the reaction of 16-bromolambertianic acid **5** with arylboronic acids **6b**,**c** in the optimized reaction conditions (*f*) afforded the cascade reaction products **7b**,**c** (yield 52–68%, entries 16,17).

Conditions (*f*) were then employed for reaction of 16-bromolambertianic acid **5** with various arylboronic acids **6e**–**j**. The reaction with 4-trifluoromethyl- and 4-cyano- substituted phenylboronic acids **6e**,**f** proceeded smoothly, preferring the formation of compounds **7e**,**f** (yield 73–81%) (Figure 2). Similar results were obtained in the reaction of bromofuran **5** with arylboronic acids having a substituent in the *meta* position of the aromatic ring-3-fluorophenyl- and 3-methoxy-phenylboronic acids **6g**,**h**. The yield of domino reaction products **7g**,**h** reaches 61–72%.

The reaction of **5** with 2-methylphenylboronic acid **6i** was carried out at 80–85 °C (conditions *e*) (Figure 2); the reaction did not proceed well at a lower temperature. The isolated yield of compound **7i** reached 48%. In these experiments, the amount of Suzuki coupling product was as low as 3–5% (from the NMR spectrum of the reaction mixture), and by subsequent column chromatography on silica gel, no fraction containing the arylation product was isolated. The reaction of diterpenoid bromide **5** with 3,4,5-trimethoxyphenylboronic acid **6j** under conditions *e* afforded a mixture of domino reaction **7j** and arylation **8j** products in 61% and 6% yields, respectively (Figure 2). When this reaction was carried out at 60–65 °C (conditions *f*), only the tetracyclic compound **7j** was obtained with the yield 67%.

The results indicated that the Pd-catalyzed interaction of 16-bromolambertianic acid **5** with arylboronic acids predominantly proceeds via the intramolecular Heck reaction followed by arylation of the resulting palladium intermediate. A high level of activity of **5** with arylboronic acid containing electron-withdrawing groups **6c**,**e**–**h** can be apparently concerning due to their higher acidity, which favors the quaternization of the boron atom at the trans-metalation stage [38]. Substitution at the ortho-position (compounds **6b**,**i**) suppressed the reactivity and the yield of 17-arylisopongian diterpenoids **7b**,**i**, and the formation of Suzuki coupling product can be caused by unfavorable stereochemical interactions both at the stage of formation of the boronate anion and during the course of intramolecular cross-coupling. Other studies on domino palladium-catalyzed Heck cyclization/Suzuki coupling have also observed direct couplings [26].

In line with those previously postulated for the Pd-catalyzed domino Heck cascade reactions [23,24], we propose a plausible mechanism (Figure 3). The oxidative addition of terpenoid bromofuran **5** to palladium(0) formed intermediate A, which was annulated through carbopalladation to form palladium(II) intermediate B. The high diastereoselectivity of the carbopalladation forming B is consistent with the previous literature: the *trans*-adduct is thought to form exclusively from the minimized axial–axial interactions [26,39]. The control over the diastereoselectivity of the formation of **7** is provided by the structural features of the labdanoid core, which consist in a higher steric accessibility of the α-side of the diterpene core. The process has been proceeding from the less hindered α-side. The trapped palladium species B reacts with the aryl boronic acid in a transmetalation step, forming intermediate **C**, which after reductive elimination afforded the desired tetracyclic compound of type **7** with the β-disposition of the C-17 aryl substituent. A direct Suzuki coupling of intermediate D led to the formation of 16-aryllambertianic acid derivatives **8**.

To study the nature of the substituent in the furan ring on the course of the above domino Heck–Suzuki reaction, we obtained several new derivatives of 16-bromolambertianic acid **5** with both electron-donating and electron-withdrawing substituents at the C-15 carbon atom of the furan ring. The key compound in the synthesis was 16-bromo-15-formyllambertianic acid **9** synthesized by the formylation of terpenoid 2-bromofuran **5** under the described Vilsmeier–Haack reaction conditions for methylambertianate [40]. Complete conversion of starting compound **5** was observed by using an 18-fold excess of phosphorus oxychloride. The isolated yield of aldehyde **9** reached to 89% (Figure 4). It should be noted that in the methods for the synthesis of furfural bromide in the literature, using furfural as the starting compound was described in [41]. The preparation of 5-bromofuran-2-carbaldehyde from bromofuran is limited by the example of formylation in DMF in the presence of butyllithium. The key compound, 16-bromo-15-cyanolambertianic acid **10** (yield 93%), was synthesized by treating aldehyde **9** with water solution of ammonia in the presence of iodine (3 equiv.) in tetrahydrofuran as described in [42] for the preparation of 16-cyanoderivatives of methyllambertianate; for comparison, the complete conversion of 16-formyl methyllambertianic acid was achieved using 1.2 equiv. of iodine [42]. The reduction in aldehyde **9** with sodium borohydride in methanol yielded 16-bromo-15-hydroxymethyllambertianic acid **11**. 16-Bromo-15-(methylaminomethyl)lambertianic acid **12** was obtained by the reductive amination of 16-bromo-15-formyllambertianic acid **9**. This transformation was carried out in three stages: treatment of methylamine hydrochloride with triethylamine, reaction of the free amine with aldehyde **9** in methylene chloride in the presence of magnesium sulfate, and subsequent reduction in the resulting imine with sodium borohydride in methanol.

Conditions (***e***) (Table 1) were then employed for the reaction of 15,16-disubstituted furanolabdanoids **9**–**10** with arylboronic acids **6a**,**f**,**j**. The reaction of compounds **9** or **10** with phenylboronic acid **6a** proceeded smoothly with the formation of Suzuki reaction coupling products **13a** or **14** (isolated yield 52–69%) (Figure 5)**.** The reaction of **9** with arylboronic acids **6f**,**j** differed in the electronic nature of the substituent, and it also gave only the product of Suzuki cross-coupling reaction **13b**,**c** (yield 49–58%). Apparently, in the reaction of furanolabdanoids with an electron-withdrawing substituent at the C-15 carbon atom of the furan ring with arylboronic acids, direct Suzuki coupling was the major pathway.

Disubstituted furanolabdanoids **11** or **12** gave no products with phenylboronic acid **6a** in the indicated conditions; the electron-donating substituent in the furan ring completely suppressed the reaction, and only the starting compounds were isolated. 

The experimental results have revealed a significant influence of the nature of the substituent in functionalized furanolabdanoid **5**,**9**–**14** on the direction of the cross-coupling/domino Heck cascade reactions. The unsuccessful reaction of arylboronic acids with 16-bromolambertianate derivatives **11** and **12**, having an electron-donating substituent in the furan ring, can be explained by the deactivating effect on the activity of the initial bromofuran in the oxidative addition step of palladium(0) (formation of intermediate A) (Figure 3). The presence of an electron-withdrawing substituent in the furan ring promotes both oxidative addition and *trans*-*cis* rearrangement of the palladium complex formed as a result of transmetalation in the Suzuki reaction. The latter effect can be explained by an increase in the *trans*-effect of furan in the palladium complex as a result of enhancement of the π-electron-withdrawing properties.

The obtained 17-aryl-isospongyan-13(16),14-diene-18-oic acids **7a**–**j** are stable in solid form during long-term storage at room temperature. When they are dissolved in chloroform, the solution (within 1–2 h) acquires a dark color.

The structure of all synthesized compounds **5**, **7a**–**j**, **8b**–**d**,**j**, **9**–**12**, **13a**–**c** and **14** was confirmed by IR, ^1^H and ^13^C spectroscopy and mass-spectrometry data. The ^1^H and ^13^C-NMR spectra of dodecahydrophenantro [1,2–*b*] furans **7a**–**j** agree with their structure and contain one set of characteristic signals of tetracyclic core and the corresponding 17-aryl substituent (Appendix A). The ^1^H-NMR spectra of tetracyclic componds **7a**–**j** were characterized by a downfield shift of the singlet signal of the C^20^H_3_ methyl group (∆δ 0.94–1.03 ppm) as compared to the signal for the corresponding proton in the spectrum of compound **5** (δ 0.57 ppm) and also of compounds **8b**–**d**,**j**,**13a**–**c** and **14** (δ 0.53–0.58 ppm). An indicative feature of the ^1^H-NMR spectra of tetracyclic compounds **7a**–**j** was the disappearance of singlet signals of protons of the exomethylene double bond and the presence of two doublets in the region of δ 2.45–3.13 ppm with coupling constants 12.6–13.9 Hz, which belong to the protons of the methylene group CH_2_-17. The (8*S*)-configuration (β-position) of the benzyl substituent was confirmed by the data of the NOESY experiments: the cross-peaks between the signal of the methyl group C^20^H_3_ and the protons H-17 (δ 2.85 and 3.13 ppm) were observed.

The mass spectra of tetracyclic compounds **7a**–**j** contain a weak peak of the molecular ion. All of the spectra were characterized with the main-ion peak with a mass of 301 (100%), which corresponded to the fragmentation of М+ with the loss of the arylmethyl substituent.

## 3. Materials and Methods

### 3.1. General Information

^1^H-NMR and ^13^C-NMR spectra were recorded by using a Bruker AV-300 (300.13 (^1^H), 75.48 MHz (^13^C)) (compounds **7g**,**j**, **8j**, **9**–**11**, **13b**), AV-400 (400.13 (^1^H), 100.78 MHz (^13^C)) (compounds **5**, **7b**–**e**,**i**, **8b**–**d**, **13c**, **14a**), DRX-500 (500.13 (^1^H), 125.77 MHz (^13^C)) (compounds **7h**, **12**, **13a**), or AV-600 (600.30 (^1^H), 150.96 MHz (^13^C)) (compounds **7a**,**f**), spectrometer. Deuterochloroform (CDCl_3_) was used as a solvent, with residual CHCl_3_ (δ_H_ = 7.24 ppm) or CDCl_3_ (δ_C_ = 77.0 ppm) being employed as internal standards. NMR signal assignments were carried out with the aid of a combination of 1D and 2D NMR techniques that included ^1^H, ^13^C, COSY, HSQC and HMBC spectra. In the description of the ^1^H-NMR spectra of compounds **5**, **7a**–**j**, **8b**–**d**,**j**, **9**–**12**, **13a**–**c** and **14**, the diterpenoid core atom numbering given in structures **1**–**3** and **4** (Figure 1) was used. IR absorption spectra were recorded on a Vector 22 FT-IR spectrometer in KВr pellets. The specific rotation values [α]_D_ were obtained on a PolAAr 3005 polarimeter. Melting points were determined using a thermosystem Mettler Toledo FP900 (Columbus, OH, USA). HRMS spectra were recorded on a Thermo Scientific DFS mass spectrometer (evaporator temperature 200–250 °C, EI ionization at 70 eV). Elemental analysis was carried out on an 1106 Elemental analysis instrument (Carlo-Erba, Milan, Italy).

The reaction progress and the purity of the obtained compounds were monitored by TLC on Silufol UV−254 plates (Kavalier, Czech Republic; CHCl_3_-EtOH, 100:1; detection under UV light or by spraying the plates with a 10% water solution of H_2_SO_4_ followed by heating at 100 °C). Products were isolated by column chromatography on silica gel 60 (0.063–0.200 mm, Merck KGaA, Darmstadt, Germany) eluting with indicated solvent systems. The chemicals used were as follows: NBS, dppf, Pd(OAc)_2_ were purchased from Acros Organics, P(Tol)_3_ and XPhos was purchased from Sigma-Aldrich (St. Louis, MO, USA); TolBINAP, P(1-Adm)_2_Bn, (*R*)-BINAP, (*S*)-BINAP and (4-cyanophenyl)boronic acid **6f** were purchased from Alfa Aesar; 3-fluorophenylboronic acid **6g** was purchased from BLD Pharm. Pd(PPh_3_)_4_, phenylboronic acids **6a**–**j** were purchased from Fluorochem. Lambertianic acid **1** was isolated from *Pinus sibirica* R. Mayr sap by the reported method [32]. [α]_D_^20^ +55.2 (*с* 2.6; EtOH). 

Solvents (DMF, CH_3_CN, CH_2_Cl_2_, CHCl_3_, EtOH, MeOH) and Et_3_N were purified by standard methods and distilled under a stream of argon just before use. Copies of NMR spectra (^1^H and ^13^C) are given in the Appendix A. Several impurities in the spectra arising from the stage of isolation of lambertianic acid **4** from plant material. 

### 3.2. Synthesis and Spectral Data of Compound (**5**), Tetracyclic Compounds (**7a**–**j**) and 16-Aryllambertianic Acid Derivatives (**8b**–**d**,**j**)

#### 3.2.1. The Reaction of Lambertianic Acid (**1**) with *N*-Bromosuccinimide

To a stirred solution of lambertianic acid **4** (1.00 g, 3.16 mmol) in CH_2_Cl_2_ (20 mL), *N*-bromosuccinimide (0.84 g, 4.75 mmol) was added portion wise at 20 °C. The mixture was stirred vigorously at room temperature for 1 h. The solvent was removed under reduced pressure, and the residue was subjected to column chromatography (chloroform). Crystallization of product fractions from hexane afforded terpenoid 2-bromofuran **5** (0.80 g, 64%) as a yellow solid. (*1S,4aR,5S,8aR*)*-*5-(2-(2-Bromofuran-3-yl)ethyl)-1,4a-dimethyl-6-methylenedecahydronaphthalene-1-carboxylic acid (16-Bromolambertianic acid) (**5**), mp 133.7 °С (decomp.); ^1^H-NMR (400 MHz, CDCl_3_, δ, ppm): 0.57 (3Н, s, СН_3_-20), 0.97 (1H, dt, *J* = 13.4, 2.4 Hz, H-1*α*), 1.01 (1H, dt, *J* = 13.6, 2.4 Hz, H-3*α*), 1.21 (3Н, s, СН_3_-19), 1.27 (1Н, m, H-5*α*), 1.48 (1Н, dm, *J* = 13.7 Hz, Н-2), 1.53–1.60 (2Н, m, Н-11, Н-9), 1.77–1.97 (6H, m, Н-11, Н-6, Н-6, Н-1*β*, Н-2, Н-7), 2.12 (1Н,dm, *J* = 13.2 Hz, Н-3*β*), 2.14–2.21 (1Н, m, Н-12), 2.40–250 (2Н, m, Н-7, Н-12), 4.61, 4.90 (2Н, both s, Н-17), 6.27 (1Н, s, Н-14), 7.35 (1Н, s, Н-15); ^13^C-NMR (125 MHz, CDCl_3_, δ, ppm): 12.75 (CH_3_-20), 19.87 (С-2), 23.76 (С-12), 23.95 (С-11), 26.02 (C-6), 28.95 (CH_3_-19), 37.84 (C-3), 38.66 (C-7), 39.00 (C-1), 40.37 (C-4), 44.21 (С-10), 55.12 (С-9), 56.29 (С-5), 106.56 (С-17), 112.99 (С-14), 119.88 (С-16), 124.12 (С-13), 143.55 (С-15), 147.59 (С-8), 184.44 (С-18); IR (KBr, ν, cm^−1^): 3435 (OH), 1691 (C=O), 1643, 1414, 879, 885 (CH=CH_2_), 1468, 1373, 1032 (furan), 621 (С-Br). Anal. Calcd for C_20_H_27_BrO_3_: С, 60.76; H, 6.88; Br, 20.21. Found: С, 60.54; H, 7.02; Br, 19.94.

#### 3.2.2. Reaction of 16-Bromolambertianic Acid (**5**) with Phenylboronic Acids (**6a**–**j**)

Conditions: (a)A mixture of 2-bromofuran **5** (0.30 g, 0.76 mmol), arylboronic acid **6a**–**c** (0.91 mmol), Pd(РPh_3_)_4_ (0.02 g, 0.02 mmol) and K_2_CO_3_ (0.38 g, 2.75 mmol) in DMF (3 mL) and H_2_O (1.5 mL) was stirred at 80–85 °C (bath) for 24 h under a stream of argon. After cooling, the stirred mixture was treated with diluted H_2_SO_4_ (0.1 mL in 1 mL of H_2_O) to pH 5 and extracted with CHCl_3_ (3 × 50 mL). The combined organic solution was washed with water (3 × 15 mL), dried over MgSO_4_, filtered and evaporated in vacuo. The residue was purified by column chromatography (eluent petroleum ether-Et_2_O, 2:1) to afford compounds **7a**, **7b** and **8b**, or **7c** and **8c**, respectively.(b)A mixture of 2-bromofuran **5** (0.30 g, 0.76 mmol), (4-methoxyphenyl)boronic acid **6d** (0.14 g, 0.91 mmol), Pd(OAc)_2_ (0.01 g, 0.04 mmol), P(Tol)_3_ (0.05 g, 0.15 mmol) and K_2_CO_3_ (0.38 g, 2.75 mmol) in DMF (3 mL) and H_2_O (1.5 mL) was stirred at 80–85 °C (bath) for 24 h under a stream of argon. After cooling, the stirred mixture was treated with diluted H_2_SO_4_ (0.1 mL in 1 mL of H_2_O) to pH 5 and extracted with CHCl_3_ (3 × 50 mL). The combined organic solution was washed with water (3 × 15 mL), dried over MgSO_4_ and evaporated in vacuo. The residue was purified by column chromatography (eluent petroleum ether-Et_2_O, 2:1) to afford compound **7d** (0.06 g, 18%).(c)A mixture of 2-bromofuran **5** (0.30 g, 0.76 mmol), (4-methoxyphenyl)boronic acid **6d** (0.14 g, 0.91 mmol), Pd(OAc)_2_ (0.01 g, 0.04 mmol), dppf (0.04 g, 0.08 mmol) and K_2_CO_3_ (0.38 g, 0.03 mmol) in DMF (3 mL) and H_2_O (1.5 mL) was stirred at 80–85 °C (bath) for 24 h under a stream of argon. After cooling, the stirred mixture was treated with diluted H_2_SO_4_ (0.1 mL in 1 mL of H_2_O) to pH 5 and extracted with CHCl_3_ (3 × 50 mL). The combined organic solution was washed with water (3 × 15 mL), dried over MgSO_4_ and evaporated in vacuo. The residue was purified by column chromatography (eluent petroleum ether-Et_2_O, 2:1) to give compounds **8d** (0.06 g, 18%) and **7d** (0.07 g, 22%).(d)A mixture of 2-bromofuran **5** (0.30 g, 0.76 mmol), (4-methoxyphenyl)boronic acid **3d** (0.14 g, 0.91 mmol), Pd(OAc)_2_ (0.01 g, 0.04 mmol), TolBINAP (0.05 g, 0.08 mmol) and K_2_CO_3_ (0.38 g, 2.75 mmol) in DMF (3 mL) and H_2_O (1.5 mL) was stirred at 80–85 °C (bath) for 24 h under a stream of argon. After cooling, the stirred mixture was treated with diluted H_2_SO_4_ (0.1 mL in 1 mL of H_2_O) to pH 5 and extracted with CHCl_3_ (3 × 50 mL). The combined organic solution was washed with water (3 × 15 mL), dried over MgSO_4_ and evaporated in vacuo. The residue was purified by column chromatography (eluent petroleum ether-Et_2_O, 2:1) to give compounds **8d** (0.04 g, 11%) and **7d** (0.07 g, 22%).(e)A mixture of 2-bromofuran **5** (0.30 g, 0.76 mmol), arylboronic acid **6d**,**i**,**j** (0.91 mmol), Pd(OAc)_2_ (0.01 g, 0.04 mmol), (*R*)-BINAP (0.05 g, 0.08 mmol) (or (*S*)-BINAP, conditions *e**), and K_2_CO_3_ (0.38 g, 2.75 mmol) in DMF (3 mL) and H_2_O (1.5 mL) was stirred at 80–85 °C (bath) for 24 h under a stream of argon. After cooling, the stirred mixture was treated with diluted H_2_SO_4_ (0.1 mL in 1 mL of H_2_O) to pH 5 and extracted with CHCl_3_ (3 × 50 mL). The combined organic solution was washed with water (3 × 15 mL), dried over MgSO_4_ and evaporated in vacuo. The residue was purified by column chromatography (eluent petroleum ether-Et_2_O, 2:1) to give compounds **7d**,**8d**,**7i**,**7j**,**8j**.(f)A mixture of 2-bromofuran **5** (0.30 g, 0.76 mmol), arylboronic acid **6b**–**h** (0.91 mmol), Pd(OAc)_2_ (0.01 g, 0.04 mmol), (*R*)-BINAP (0.05 g, 0.08 mmol) and K_2_CO_3_ (0.38 g, 2.75 mmol) in DMF (3 mL) and H_2_O (1.5 mL) was stirred at 60–65 °C (bath) for 24 h under a stream of argon. After cooling, the stirred mixture was treated with diluted H_2_SO_4_ (0.1 mL in 1 mL of H_2_O) to pH 5 and extracted with CHCl_3_ (3 × 50 mL). The combined organic solution was washed with water (3 × 15 mL), dried over MgSO_4_ and evaporated in vacuo. The residue was purified by column chromatography (eluent petroleum ether-Et_2_O, 2:1) to afford compound 7**b**,**7c**,**7e**–**j**,**7d**,**8d**.(g)A mixture of 2-bromofuran **5** (0.30 g, 0.76 mmol), (4-methoxyphenyl)boronic acid **6d** (0.14g, 0.91 mmol), Pd(OAc)_2_ (0.01 g, 0.04 mmol), (*R*)-BINAP (0.05 g, 0.08 mmol) and Ce_2_CO_3_ (0.89 g, 2.74 mmol) in DMF (3 mL) and H_2_O (1.5 mL) was stirred at 60–65 °C (bath) for 24 h under a stream of argon. After cooling, the stirred mixture was treated with diluted H_2_SO_4_ (0.1 mL in 1 mL of H_2_O) to pH 5 and extracted with CHCl_3_ (3 × 50 mL). The combined organic solution was washed with water (3 × 15 mL), dried over MgSO_4_ and evaporated in vacuo. The residue was purified by column chromatography (eluent petroleum ether-Et_2_O, 2:1) to give compounds **7d**,**8d**.(h)A mixture of 2-bromofuran **5** (0.30 g, 0.76 mmol), (4-methoxyphenyl)boronic acid **6d** (0.14 g, 0.91 mmol), Pd(OAc)_2_ (0.01 g, 0.04 mmol), (*R*)-BINAP (0.05 g, 0.08 mmol) and K_2_CO_3_ (0.38 g, 2.75 mmol) in DMF (3 mL) and H_2_O (1.5 mL) was stirred at 40–45 °C (bath) for 24 h under a stream of argon. After cooling, the stirred mixture was treated with diluted H_2_SO_4_ (0.1 mL in 1 mL of H_2_O) to pH 5 and extracted with CHCl_3_ (3 × 50 mL). The organic layer was washed with water (3 × 15 mL), dried over MgSO_4_ and evaporated in vacuo. The residue was purified by column chromatography (eluent petroleum ether-Et_2_O, 2:1) to give unreacted 2-bromofuran ***5*** (0.21 g), compounds **7d** (0.02 g, 6%) and **4d** (0.02 g, 6%).(i)A mixture of 2-bromofuran **5** (0.30 g, 0.76 mmol), (4-methoxyphenyl)boronic acid **6d** (0.14g, 0.91 mmol), Pd(OAc)_2_ (0.01 g, 0.04 mmol), (*R*)-BINAP (0.05 g, 0.08 mmol) and K_2_CO_3_ (0.38 g, 2.75 mmol) in CH_3_CN (3 mL) and H_2_O (1.5 mL) was stirred at 60–65 °C (bath) for 24 h under a stream of argon. After cooling, the stirred mixture was treated with diluted H_2_SO_4_ (0.1 mL in 1 mL of H_2_O) to pH 5 and extracted with CHCl_3_ (3 × 50 mL). The combined organic solution was washed with water (3 × 15 mL), dried over MgSO_4_ and evaporated in vacuo. The residue was purified by column chromatography (eluent petroleum ether-Et_2_O, 2:1) to give compounds **2** (0.13 g) and 7**d** (0.08 g, 24%).

#### 3.2.3. Spectral Data of Tetracyclic Compounds (**7a**–**j**) and 16-Aryllambertianic Acid Derivatives (**8b**–**d**,**j**)

(3b*S*,5a*R*,6*S*,9a*R*,9b*R*)-3b-Benzyl-6,9a-dimethyl-3b,4,5,5a,6,7,8,9,9a,9b,10,11-dodecahydro-phenanthro [1,2–*b*]furan-6-carboxylic acid (**7a**). Yield 71% (a). White needles. m.p. 154 °С (decomp.); [α]_D_^25^ −7.32 (*c* 0.4, EtOH). ^1^H-NMR (600 MHz, CDCl_3_, δ, ppm): 0.96 (1H, dt, *J* = 12.8, 3.7 Hz, H-1*α*), 1.00 (3Н, s, СН_3_-20), 1.07 (1H, dt, *J* = 13.6, 4.4 Hz, H-3*α*), 1.25 (1Н, dd, *J* = 13.2, 2.2 Hz, H-5*α*), 1.27 (1Н, dm, *J* = 5.1 Hz, H-7), 1.31 (3Н, s, СН_3_-19), 1.44 (1Н, d, *J* = 11.7 Hz, Н-9), 1.52 (1Н, dm, *J* = 13.6 Hz, Н-2), 1.80–2.03 (5H, m, Н-11,6, 11,1*β*,2), 2.21 (1Н, dm, *J* = 13.6 Hz, Н-3*β*), 2.31 (1Н, ddd, *J* = 18.0, 16.1, 2.9 Hz, Н-12), 2.39–2.44 (1Н, m, Н-6), 2.45 (1Н, dm, *J* = 13.9 Hz, Н-7), 2.62 (1Н, dd, *J* = 16.1, 5.9 Hz, Н-12), 2.84 (1Н, d, *J* = 13.2 Hz, Н-17), 3.12 (1Н, d, *J* = 13.2 Hz, Н-17), 6.08 (1Н, d, *J* = 1.5 Hz, Н-14), 6.77–6.78 (2Н, m, 2Н-Ph), 6.87 (1Н, d, *J* = 1.5 Hz, Н-15), 7.15–7.16 (3Н, m, 3Н-Ph); ^13^C-NMR (150 MHz, CDCl_3_, δ, ppm): 14.39 (CH_3_-20), 17.96 (С-11), 18.98 (C-6), 19.29 (С-2), 22.42 (С-12), 28.85 (CH_3_-19), 33.08 (C-7), 37.83 (C-3)*, 37.94 (C-1)*, 40.07 (C-17)**, 40.30 (C-8)**, 41.71 (С-10), 43.87 (C-4), 57.29 (С-9), 57.41 (С-5), 106.95 (С-14), 114.82 (С-13), 125.62 (С-Ph), 127.38 (2С-Ph), 130.20 (2С-Ph), 139.58 (С-Ph), 139.84 (С-15), 156.65 (С-16), 184.71 (С-18); UV (EtOH) λ_max_, (lgε): 256 (2.85) nm. IR (KBr, ν, cm^−1^): 3429 (OH), 1695 (C=O), 1497, 1468, 729, 698 (Ph); 1452, 1373, 1032 (furan); HR-MS, calcd. for C_26_H_32_O_3_: 392.2338 found, [M]^+^ *m*/*z*: 392.2346.

(3b*S*,5a*R*,6*S*,9a*R*,9b*R)-*3b-(2-Fluorobenzyl)-6,9a-dimethyl-3b,4,5,5a,6,7,8,9,9a,9b,10,11-dodecahydrophenanthro [1,2–*b*]furan-6-carboxylic acid (**7b**). Yield: 23% (a), 52% (f). White solid. m.p. 145 °С (decomp.); [α]_D_^25^ −20.0 (*c* 0.05, EtOH). ^1^H-NMR (400 MHz, CDCl_3_, δ, ppm): 0.93 (1H, dt, *J* = 12.2, 5.7 Hz, H-1*α*), 0.96 (3Н, s, СН_3_-20), 1.03 (1H, dt, *J* = 13.4, 4.3 Hz, H-3*α*), 1.19–1.24 (2Н, m, H-5*α*, H-7), 1.27 (3Н, s, СН_3_-19), 1.41 (1Н, d, *J* = 10.6 Hz, Н-9), 1.49 (1Н, dm, *J* = 14.2 Hz, Н-2), 1.79–2.00 (5H, m, Н-11,6,11,1*β*,2), 2.17 (1Н, dm, *J* = 13.6 Hz, Н-3*β*), 2.25–2.38 (2Н, m, Н-12,6), 2.41 (1Н, m, Н-7), 2.60 (1Н, dd, *J* = 16.0, 5.6 Hz, Н-12), 2.81 (1Н, d, *J* = 12.9 Hz, Н-17), 3.17 (1Н, d, *J* = 12.9 Hz, Н-17), 6.04 (1Н, d, *J* = 1.8 Hz, Н-14), 6.68 (1Н, *J* = 7.7 Hz, t, Н-5′), 6.81 (1Н, d, *J* = 1.8 Hz, Н-15), 6.86 (1Н, t, *J* = 7.7 Hz, Н-4′), 6.89 (1Н, m, Н-6′), 7.09 (1Н, m, Н-3′); ^13^C-NMR (75 MHz, CDCl_3_, δ, ppm): 14.33 (CH_3_-20), 18.09 (С-11), 18.97 (C-6), 19.30 (С-2), 22.52 (С-12), 28.83 (CH_3_-19), 32.89 (C-7), 33.51 (C-17), 37.92 (C-3, C-1), 40.01 (C-8), 41.81 (С-10), 43.81 (C-4), 57.32 (С-9), 57.41 (С-5), 109.65 (С-14), 115.03 (С-13), 114.61 (d, ^2^*J*_CF_ = 23.2 Hz, С-3′), 123.11 (С-5′), 126.65 (d, ^2^*J*_CF_ = 16.0 Hz, С-1′), 127.47 (d, ^3^*J*_CF_ = 8.3 Hz, С-4′), 132.48 (С-6′), 139.97 (С-15), 156.54 (С-16), 161.57 (d, ^1^*J*_CF_ = 244.6 Hz, С-2′), 183.63 (С-18); UV (EtOH) λ_max_, (lgε): 263 (3.47), 270 (3.42) nm; IR (KBr, ν, cm^−1^): 3400 (ОН), 1695 (C=O); 1452, 1373, 1038 (furan); 1583, 1491, 758 (Ar); 1228 (C-F); HR-MS, calcd. for C_26_H_31_O_3_F: 410.2252; found, [M]^+^ *m*/*z*: 410.2247.

(1*S*,4a*R*,5*S*,8a*R)-*5-(2-(2-(2-Fluorophenyl)furan-3-yl)ethyl)-1,4a-dimethyl-6-methylenedecahydronaphthalene-1-carboxylic acid (**8b**). Yield 45% (a). White solid. m.p. 52.1 °С (decomp.); [α]_D_^25^ +10.8 (*c* 0.1, EtOH). ^1^H-NMR (400 MHz, CDCl_3_, δ, ppm): 0.55 (3Н, s, СН_3_-20), 0.96 (1H, dm, *J* = 3.8 Hz, H-1*α*), 1.00 (1H, dt, *J* = 13.4, 3.8 Hz, H-3*α*), 1.20 (3Н, s, СН_3_-19), 1.25 (1Н, m, H-5*α*), 1.46 (1Н, m, Н-2), 1.55–1.63 (2Н, m, Н-11,9), 1.69–1.94 (m, 6H, Н-6, 11,6,7,1*β*,2), 2.10 (1Н, dm, *J* = 13.3 Hz, Н-3*β*), 2.26–2.41 (1Н, m, Н-12), 2.33 (1Н, dm, *J* = 10.9 Hz, Н-7), 2.61 (1Н, m, Н-12), 4.50 (1Н, s, Н-17), 4.79 (1Н, s, Н-17), 6.38 (1Н, d, *J* = 1.7 Hz, Н-14), 7.10 (1Н, m, Н-4′), 7.15 (1Н, t, *J* = 7.5 Hz, Н-6′), 7.29 (m, 1Н, Н-3′), 7.41 (dt, 1Н, *J* = 7.5, 1.5 Hz, Н-5′); ^13^C-NMR (125 MHz, CDCl_3_, δ, ppm): 12.72 (CH_3_-20), 19.80 (С-2), 24.02 (С-12), 24.24 (С-11), 25.95 (C-6), 28.95 (CH_3_-19), 37.80 (C-3), 38.53 (C-7), 38.83 (C-1), 40.36 (C-10), 44.10 (С-4), 55.31 (С-9), 56.15 (С-5), 106.39 (С-17), 112.61 (С-14), 116.52 (d, ^2^*J*_CF_ = 22.2 Hz, С-3′), 119.55 (d, ^2^*J*_CF_ = 14.3 Hz, С-1′), 123.91 (С-13), 123.94 (С-5′), 129.43 (d, ^3^*J*_CF_ = 8.1 Hz, С-4′), 130.33 (d, ^3^*J*_CF_ = 3.0 Hz, С-6′), 142.24 (С-15), 143.91 (С-16), 147.72 (С-8), 159.22 (d, ^1^*J*_CF_ = 249.5 Hz, С-2′), 183.98 (С-18); UV (EtOH) λ_max_, (lgε): 264 (3.91) nm; IR (KBr, ν, cm^−1^): 3400 (ОН), 1691 (C=O); 1452, 1408, 1038 (furan); 1581, 1489, 758 (Ar); 1219 (C-F); HR-MS, calcd. for C_26_H_31_O_3_F: 410.2252; found, [M]^+^ *m*/*z*: 410.2253.

(3b*S*,5a*R*,6*S*,9a*R*,9b*R*)-3b-(4-Fluorobenzyl)-6,9a-dimethyl-3b,4,5,5a,6,7,8,9,9a,9b,10,11-dodecahydro-phenanthro [1,2*–b*]furan-6-carboxylic acid (**7c**). Yield: 40% (a), 68% (f). White solid. m.p. 102–105 °С; [α]_D_^25^ +2.9 (*c* 0.5, EtOH). ^1^H-NMR (400 MHz, CDCl_3_, δ, ppm): 0.93–0.95 (1H, m, H-1*α*), 0.95 (3Н, s, СН_3_-20), 1.04 (1H, dt, *J* = 13.5, 4.2 Hz, H-3*α*), 1.21–1.24 (2Н, m, H-5*α*,7), 1.28 (3Н, s, СН_3_-19), 1.41 (1Н, d, *J* = 11.4 Hz, Н-9), 1.49 (1Н, dm, *J* = 14.2 Hz, Н-2), 1.71–2.00 (5H, m, Н-11,6,11,1*β*,2), 2.16–2.30 (1Н, m, Н-12), 2.18 (1Н, dm, *J* = 13.5 Hz, Н-3*β*), 2.35–2.43 (2Н, m, Н-7,6), 2.59 (1Н, dd, *J* = 16.1, 6.1 Hz, Н-12), 2.77 (1Н, d, *J* = 13.0 Hz, Н-17), 3.06 (1Н, d, *J* = 13.0 Hz, Н-17), 6.06 (d, *J* = 1.7 Hz, Н-14), 6.66 (d, *J* = 8.6 Hz, 1Н, Н-2′), 6.67 (d, *J* = 8.6 Hz, 1Н, Н-6′), 6.81 (1Н, t, *J* = 8.6 Hz, 2Н, Н-3′,5′), 6.85 (d, 1Н, *J* = 1.7 Hz, Н-15); ^13^C-NMR (125 MHz, CDCl_3_, δ, ppm): 14.38 (CH_3_-20), 17.92 (С-11), 18.93 (C-6), 19.21 (С-2), 22.39 (С-12), 28.84 (CH_3_-19), 32.89 (C-7), 37.79 (C-3)*, 37.88 (C-1)*, 39.38 (C-17)**, 39.99 (C-8)**, 41.60 (С-10), 43.83 (C-4), 57.12 (С-9), 57.29 (С-5), 109.73 (С-14), 114.18 (d, ^2^*J*_CF_ = 21.0 Hz, С-3′,5′), 114.99 (С-13), 131.40 (d, ^3^*J*_CF_ = 7.6 Hz, С-2′,6′), 135.19 (d, ^4^*J*_CF_ = 3.0 Hz, С-1′), 139.95 (С-15), 156.38 (С-16), 161.37 (d, ^1^*J*_CF_ = 243.2 Hz, С-4′), 184.54 (С-18); UV (EtOH) λ_max_, (lgε): 267 (3.51), 273 (3.48) nm; IR (KBr, ν, cm^−1^): 3400 (OH), 1695 (C=O); 1414, 1034 (furan); 1603, 1488, 1450, 756 (Ar); 1223 (C-F); HR-MS, calcd. for C_19_H_25_O_3_ [М+ - (СH_2_C_6_H_4_-F)]: 301.1798; found, [М^+^-(СH_2_C_6_H_4_-F)] *m*/*z*: 301.1796.

(1*S*,4a*R*,5*S*,8a*R*)-5-(2-(2-(4-Fluorophenyl)furan-3-yl)ethyl)-1,4a-dimethyl-6-methylenedecahydro-naphthalene-1-carboxylic acid (**8с**). Yield 20% (a). Colorless oil; [α]_D_^25^ −24.0 (*c* 0.25, EtOH). ^1^H-NMR (400 MHz, CDCl_3_, δ, ppm): 0.58 (3Н, s, СН_3_-20), 0.97 (1H, m, H-1*α*), 1.00 (1H, dt, *J* = 13.4, 4.0 Hz, H-3*α*), 1.21 (3Н, s, СН_3_-19), 1.27 (1Н, m, H-5*α*), 1.47 (1Н, m, Н-2), 1.59–1.70 (2Н, m, Н-11,9), 1.77–1.96 (6H, m, Н-6,11,6,7,1*β*,2), 2.11 (1Н, dm, *J* = 12.9 Hz, Н-3*β*), 2.40 (1Н, dm, *J* = 6.2 Hz, Н-7), 2.44–2.50 (1Н, m, Н-12), 2.72 (1Н, m, Н-12), 4.59 (1Н, s, Н-17), 4.89 (1Н, s, Н-17), 6.34 (1Н, d, *J* = 1.5 Hz, Н-14), 7.04-7.07 (2Н, m, *J* = 8.7 Hz, Н-3′,5′), 7.36 (1Н, d, *J* = 1.5 Hz, Н-15), 7.51–7.53 (1Н, m, *J* = 8.7 Hz, Н-2′,6′); ^13^C-NMR (75 MHz, CDCl_3_, δ, ppm): 12.79 (CH_3_-20), 19.83 (С-2), 24.14 (С-12), 24.33 (С-11), 25.99 (C-6), 28.95 (CH_3_-19), 37.83 (C-3), 38.59 (C-7), 38.95 (C-1), 40.44 (C-10), 44.14 (С-4), 55.17 (С-9), 56.18 (С-5), 106.61 (С-17), 113.28 (С-14), 115.42 (d, ^2^*J*_CF_ = 21.6 Hz, С-3′,5′), 121.19 (С-13), 127.55 (d, ^4^*J*_CF_ = 3.3 Hz, С-1′), 127.40 (d, ^3^*J*_CF_ = 8.0 Hz, С-2′,6′), 140.97 (С-15), 147.83 (С-16), 147.87 (С-8), 161.71 (d, ^1^*J*_CF_ = 246.9 Hz, С-4′), 183.66 (С-18); UV (EtOH) λ_max_, (lgε): 259 (3.70) nm; IR (KBr, ν, cm^−1^): 3408 (OH), 1693 (C=O), 1450, 1385, 1032 (furan), 1601, 1499, 756 (Ar), 1234 (C-F); HR-MS, calcd. for C_26_H_31_O_3_F: 410.2252; found, [M]^+^ *m*/*z*: 410.2258.

(3b*S*,5a*R*,6*S*,9a*R*,9b*R*)-3b-(4-Methoxybenzyl)-6,9a-dimethyl-3b,4,5,5a,6,7,8,9,9a,9b,10,11-dodecahydrophenanthro [1,2–*b*]furan-6-carboxylic acid (**7d**). Yield: 18% (b), 22% (c), 22% (d), 38% (e), 35% (e***), 56% (f), 35% (g), 6% (h), 24% (i). White solid. m.p. 188.6 °С (decomp.); [α]_D_^25^ −50.4 (*c* 1.0, EtOH); ^1^H-NMR (400 MHz, CDCl_3_, δ, ppm): 0.92 (1H, dt, *J* = 13.7, 4.2 Hz, H-1*α*), 0.95 (3Н, s, СН_3_-20), 1.03 (1H, dt, *J* = 13.5, 3.8 Hz, H-3*α*), 1.20–1.23 (2Н, m, H-5*α*, H-7), 1.27 (3Н, s, СН_3_-19), 1.40 (1Н, d, *J* = 11.6 Hz, Н-9), 1.48 (1Н, dm, *J* = 14.2 Hz, Н-2), 1.77–1.96 (5H, m, Н-11,6,11, 1*β*,2), 2.17 (1Н, dm, *J* = 13.4 Hz, Н-3*β*), 2.15–2.30 (1Н, m, Н-12), 2.35–2.42 (2Н, m, Н-6, Н-7), 2.58 (1Н, dd, *J* = 15.9, 5.7 Hz, Н-12), 2.75 (1Н, d, *J* = 13.1 Hz, Н-17), 3.03 (1Н, d, *J* = 13.1 Hz, Н-17), 3.75 (3Н, s, ОСН_3_), 6.06 (1Н, s, Н-14), 6.63 (2Н, d, *J* = 8.4 Hz, Н-3′,5′)*, 6.68 (2Н, d, *J* = 8.4 Hz, Н-2′,6′), 6.88 (s, 1Н, Н-15); ^13^C-NMR (75 MHz, CDCl_3_, δ, ppm): 14.43 (CH_3_-20), 17.92 (С-11), 18.98 (C-6), 19.28 (С-2), 22.44 (С-12), 28.86 (CH_3_-19), 32.96 (C-7), 37.87 (C-3)*, 37.92 (C-1)*, 39.33 (C-17)**, 40.07 (C-8)**, 41.66 (С-10), 43.82 (C-4), 55.08 (ОСН_3_), 57.19 (С-9), 57.37 (С-5), 109.68 (С-14), 112.84 (С-3′,5′), 114.79 (С-13), 131.05 (С-2′, 6′), 131.60 (С-1′), 139.88 (С-15), 156.84 (С-16), 157.69 (С-4′), 183.88 (С-18); UV (EtOH) λ_max_, (lgε): 276 (4.11) nm; IR (KBr, ν, cm^−1^): 3421 (OH), 1692 (C=O), 1452, 1040 (furan), 1574, 831, 735 (Ar), 1250, 1038 (С-О); HR-MS, calcd. for C_27_H_34_O_4_: 422.2452; found, [M]^+^ *m*/*z*: 422.2454.

(1*S*,4a*R*,5*S*,8a*R*)-5-(2-(2-(4-Methoxyphenyl)furan-3-yl)ethyl)-1,4a-dimethyl-6-methylenedecahydronaphthalene-1-carboxylic acid (***8d***). Yield: 18% (c), 11% (d), 38% (e), 8% (e***), 28% (f), 21% (g), 6% (h). White solid. m.p. 124.6 °С (decomp.); [α]_D_^25^ +17.7 (*c* 0.3, EtOH). ^1^H-NMR (400 MHz, CDCl_3_, δ, ppm): 0.58 (3Н, s, СН_3_-20), 0.98 (1H, dt, *J* = 13.8, 3.5 Hz, H-1*α*), 1.01 (1H, dt, *J* = 13.4, 4.0 Hz, H-3*α*), 1.20 (3Н, s, СН_3_-19), 1.28 (1Н, m, H-5*α*), 1.47 (1Н, m, Н-2), 1.65 (2Н, m, Н-11, H-9), 1.78–1.96 (6H, m, Н-6,11,6,7,1*β*,2), 2.11 (1Н, dm, *J* = 12.9 Hz, Н-3*β*), 2.40 (1Н, dm, *J* = 7.9 Hz, Н-7), 2.43–2.51 (1Н, m, Н-12), 2.72 (1Н, m, Н-12), 3.82 (3Н, s, ОСН_3_), 4.60 (1Н, s, Н-17), 4.89 (1Н, s, Н-17), 6.33 (1Н, d, *J* = 1.8 Hz, Н-14), 6.90 (2Н, d, *J* = 8.9 Hz, Н-3′,5′), 7.34 (d, 1Н, *J* = 1.8 Hz, Н-15), 7.49 (d, *J* = 8.9 Hz, 2Н, Н-2′,6′); ^13^C-NMR (75 MHz, CDCl_3_, δ, ppm): 12.77 (CH_3_-20), 19.83 (С-2), 24.21 (С-12), 24.35 (С-11), 25.98 (C-6), 28.94 (CH_3_-19), 37.79 (C-3), 38.59 (C-7), 38.92 (C-1), 40.42 (C-4), 44.13 (С-10), 55.20 (С-9), 55.27 (ОСН_3_), 56.15 (С-5), 106.60 (С-17), 113.15 (С-14), 113.86 (С-3′,5′), 120.05 (С-13), 127.06 (С-2′,6′), 124.67 (С-1′), 140.40 (С-15), 147.89 (С-8), 148.60 (С-16), 158.48 (С-4′), 184.00 (С-18); UV (EtOH) λ_max_, (lgε): 274 (3.87) nm; IR (KBr, ν, cm^−1^): 3400 (OH), 1693 (C=O), 1452, 1385, 1252, 1032 (furan, С-О), 1574, 1466, 833, 756 (Ar); HR-MS, calcd. for C_27_H_34_O_4_: 422.2452; found, [M]^+^ *m*/*z*: 422.2457.

(5a*R*,6*S*,9a*R*,9b*R*)-6,9a-Dimethyl-3b-(4-(Trifluoromethyl)benzyl)-3b,4,5,5a,6,7,8,9,9a,9b,10,11-dodecahydrophenanthro [1,2-*b*]furan-6-carboxylic acid (**7e**). Yield 81% (f). White solid. m.p. 134.6 °С (decomp.); [α]_D_^25^ +20.8 (*c* 0.5, EtOH). ^1^H-NMR (400 MHz, CDCl_3_, δ, ppm): 0.94–0.98 (1H, m, H-1*α*), 0.96 (3Н, s,СН_3_-20), 1.05 (1H, dt, *J* = 13.4, 4.2 Hz, H-3*α*), 1.21–1.24 (2Н, m, H-5*α*,7), 1.29 (3Н, s, СН_3_-19), 1.43 (1Н, d, *J* = 11.7 Hz, Н-9), 1.50 (1Н, dm, *J* = 12.1 Hz, Н-2), 1.72–2.01 (5H, m, Н-11,6,11,1*β*,2), 2.19 (1Н, dm, *J* = 13.4 Hz, Н-3*β*), 2.27 (1Н, m, Н-12), 2.34–2.44 (1Н, m, Н-6), 2.36 (1Н, dm, *J* = 13.2 Hz, Н-7), 2.60 (1Н, dd, *J* = 16.1, 6.0 Hz, Н-12), 2.85 (1Н, d, *J* = 12.6 Hz, Н-17), 3.13 (d, 1Н, *J* = 12.6 Hz, Н-17), 6.06 (1Н, d, *J* = 1.7 Hz, Н-14), 6.81 (1Н, d, *J* = 1.7 Hz, Н-15), 6.83 (2Н, d, *J* = 8.0 Hz, С-2′,6′), 7.37 (2Н, d, *J* = 8.0 Hz, С-3′,5′); ^13^C-NMR (125 MHz, CDCl_3_, δ, ppm): 14.37 (CH_3_-20), 17.99 (С-11), 18.90 (C-6), 19.22 (С-2), 22.37 (С-12), 28.85 (CH_3_-19), 32.97 (C-7), 37.77 (C-3)*, 37.88 (C-1)*, 39.96 (C-17)**, 40.17 (C-8)**, 41.68 (С-10), 43.83 (C-4), 57.15 (С-9), 57.22 (С-5), 109.79 (С-14), 115.23 (С-13), 124.23 (d, ^3^*J*_CF_ = 3.7 Hz, С-3′,5′), 125.45 (q, ^1^*J*_CF_ = 272 Hz, СF_3_), 127.90 (d, ^2^*J*_CF_ = 32.1 Hz, С-4′), 130.36 (С-2′,6′), 140.00 (С-15), 143.87 (С-1′), 155.93 (С-16), 184.52 (С-18); UV (EtOH) λ_max_, (lgε): 216 (4.07), 280 (3.31) nm; IR (KBr, ν, cm^−1^): 3400 (OH), 1695 (C=O); 1670, 1468, 1450, 837, 756 (Ar); 1222, 1157 (CF_3_); HR-MS, calcd. for C_27_H_31_O_3_F: 460.2220; found, [M]^+^ *m*/*z*: 460.2214.

(3b*S*,5a*R*,6*S*,9a*R*,9b*R*)-3b-(4-Cyanobenzyl)-6,9a-dimethyl-3b,4,5,5a,6,7,8,9,9a,9b,10,11-dodecahydrophenanthro [1,2-*b*]furan-6-carboxylic acid (**7f**). Yield 73% (f). White solid. m.p. 119–121 °С; [α]_D_^25^ −33.5 (*c* 0.4, EtOH). ^1^H-NMR (600 MHz, CDCl_3_, δ, ppm): 0.92–0.95 (1H, m, H-1*α*), 0.95 (3Н, s, СН_3_-20), 1.04 (1H, dt, *J* = 13.6, 4.2 Hz, H-3*α*), 1.22 (1H, dd, *J* = 12.5, 1.8 Hz, H-5*α*), 1.28 (4Н, s, СН_3_-19, H-7), 1.42 (1Н, d, *J* = 11.9 Hz, Н-9), 1.50 (1Н, dm, *J* = 14.1 Hz, Н-2), 1.74 (1Н, m, Н-11), 1.86 (1Н, d, *J* = 12.9 Hz, Н-1*β*), 1.91–1.96 (2H, m, Н-11, Н-2), 2.00 (1Н, dd, *J* = 14.3, 2.6 Hz, Н-6), 2.18 (1Н, dm, *J* = 13.3 Hz, Н-3*β*), 2.24 (1Н, ddd, *J* = 17.5, 16.1, 2.4 Hz, Н-12), 2.32 (1Н, dm, *J* = 13.7 Hz, Н-7), 2.39 (1Н, m, Н-6), 2.59 (1Н, dd, *J* = 16.1, 6.3 Hz, Н-12), 2.85 (1Н, d, *J* = 12.5 Hz, Н-17), 3.13 (1Н, d, *J* = 12.5 Hz, Н-17), 6.04 (1Н, d, *J* = 1.6 Hz, Н-14), 6.79 (1Н, d, *J* = 1.6 Hz, Н-15), 6.82 (2Н, d, *J* = 8.1 Hz, С-2′,6′), 7.40 (d, 2Н, *J* = 8.1 Hz, С-3′,5′); ^13^C-NMR (150 MHz, CDCl_3_, δ, ppm): 14.37 (CH_3_-20), 18.02 (С-11), 18.91 (C-6), 19.21 (С-2), 22.33 (С-12), 28.83 (CH_3_-19), 33.08 (C-7), 37.78 (C-3)*, 37.89 (C-1)*, 39.95 (C-17)**, 40.59 (C-8)**, 41.84 (С-10), 43.84 (C-4), 57.19 (С-9), 57.20 (С-5), 109.48 (С-14), 109.87 (С-4′), 115.43 (С-13), 119.28 (C≡N), 130.83 (С-2′,6′)*, 131.16 (С-3′,5′)*, 140.09 (С-15), 145.63 (С-1′), 155.60 (С-16), 184.13 (С-18); UV (EtOH) λ_max_, (lgε): 234 (4.18) nm; IR (KBr, ν, cm^−1^): 3431 (OH), 1693 (C=O), 2227 (C≡N), 1468, 1450, 754 (Ar); HR-MS, calcd. for C_27_H_31_O_3_N: 417.2299; found, [M]^+^ *m*/*z*: 417.2297.

(3b*S*,5a*R*,6*S*,9a*R*,9b*R*)-3b-(3-Fluorobenzyl)-6,9a-dimethyl-3b,4,5,5a,6,7,8,9,9a,9b,10,11-dodecahydrophenanthro [1,2–*b*]furan-6-carboxylic acid (**7g**). Yield 61% (f). White solid. m.p. 101 °С (decomp.); [α]_D_^25^ +0.6 (*c* 0.35, EtOH); ^1^H-NMR (300 MHz, CDCl_3_, δ, ppm): 0.92–0.95 (1H, m, H-1*α*), 0.95 (s, 3Н, СН_3_-20), 1.03 (1H, dt, *J* = 13.0, 3.5 Hz, H-3*α*), 1.23 (2H, m, H-5*α*,7), 1.28 (3Н, s, СН_3_-19), 1.41 (1Н, d, *J* = 11.4 Hz, Н-9), 1.49 (dm, 1Н, *J* = 14.7 Hz, Н-2), 1.73–2.00 (5Н, m, Н-11,6,11,1*β*,2), 2.16–2.22 (2Н, m, Н-3*β,*12), 2.40 (2Н, m, Н-7,6), 2.59 (1Н, dd, *J* = 16.0, 6.0 Hz, Н-12), 2.79 (1Н, d, *J* = 12.8 Hz, Н-17), 3.07 (1Н, d, *J* = 12.8 Hz, Н-17), 6.07 (1Н, d, *J* = 1.7 Hz, Н-14), 6.39 (1Н, d, *J* = 11.0 Hz, С-2′), 6.55 (1Н, d, *J* = 7.6 Hz, С-6′), 6.86 (1Н, d, *J* = 1.7 Hz, Н-15), 6.83 (1Н, t, *J* = 7.6 Hz, C-5′), 7.08 (1Н, dd, *J* = 14.2, 7.6 Hz, С-4′); ^13^C-NMR (125 MHz, CDCl_3_, δ, ppm): 14.36 (CH_3_-20), 17.93 (С-11), 18.91 (C-6), 19.19 (С-2), 22.36 (С-12), 28.83 (CH_3_-19), 32.99 (C-7), 37.76 (C-3)*, 37.87 (C-1)*, 39.97 (C-17)**, 40.02 (C-8)**, 41.63 (С-10), 43.81 (C-4), 57.14 (С-9), 57.26 (С-5), 109.78 (С-14), 112.49 (d, ^2^*J*_CF_ = 21.0 Hz, С-4′), 115.07 (С-13), 116.91 (d, ^2^*J*_CF_ = 20.8 Hz, С-2′), 125.87 (С-6′), 128.56 (d, ^3^*J*_CF_ = 8.3 Hz, С-5′), 139.97 (С-15), 142.22 (d, ^3^*J*_CF_ = 7.4 Hz, С-1′), 156.15 (С-16), 162.33 (d, ^1^*J*_CF_ = 244.1 Hz, С-3′), 184.48 (С-18); UV (EtOH) λ_max_, (lgε): 263 (3.56), 270 (3.50) nm; IR (KBr, ν, cm^−1^): 3429 (OH), 1693 (C=O), 1587, 1410, 1047 (furan), 1595, 1487, 1448, 756 (Ar), 1254 (C-F); HR-MS, calcd. for C_26_H_31_O_3_F: 410.2252; found, [M]^+^ *m*/*z*: 410.2251.

(3b*S*,5a*R*,6*S*,9a*R*,9b*R*)-3b-(3-Methoxybenzyl)-6,9a-dimethyl-3b,4,5,5a,6,7,8,9,9a,9b,10,11-dodecahydrophenanthro [1,2–*b*]furan-6-carboxylic acid (**7h**). Yield 72% (f). White solid. m.p. 193 °С (decomp.); [α]_D_^25^ −8.6 (*c* 0.2, EtOH); ^1^H-NMR (500 MHz, CDCl_3_, δ, ppm): 0.93 (1H, dt, *J* = 13.2, 3.5 Hz, H-1*α*), 0.96 (3Н, s, СН_3_-20), 1.04 (1H, dt, *J* = 13.5, 4.2 Hz, H-3*α*), 1.22 (1H, dd, *J* = 13.2, 2.0 Hz, H-5*α*), 1.24 (1Н, m, Н-7), 1.27 (3Н, s, СН_3_-19), 1.41 (1Н, d, *J* = 11.2 Hz, Н-9), 1.48 (1Н, dm, *J* = 14.4 Hz, Н-2), 1.78–1.99 (5Н, m, Н-11,6,11,1*β*,2), 2.17 (1Н, dm, *J* = 13.4 Hz, Н-3*β*), 2.26 (1Н, ddd, *J* = 17.3, 13.6, 2.5 Hz, Н-12), 2.35–2.41 (1Н, m, Н-6), 2.43 (1Н, dm, *J* = 13.2 Hz, Н-7), 2.59 (1Н, dd, *J* = 16.1, 6.0 Hz, Н-12), 2.78 (1Н, d, *J* = 12.7 Hz, Н-17), 3.06 (1Н, d, *J* = 12.7 Hz, Н-17), 3.82 (3Н, s, ОСН_3_), 6.07 (1Н, d, *J* = 1.8 Hz, Н-14), 6.17 (1Н, d, *J* = 2.3 Hz, С-2′), 6.44 (1Н, d, *J* = 7.7 Hz, С-6′), 6.69 (1Н, dd, *J* = 7.7, 2.3 Hz, С-4′), 6.88 (1Н, d, *J* = 1.8 Hz, Н-15), 7.06 (1Н, t, *J* = 7.7 Hz, C-5′); ^13^C-NMR (125 MHz, CDCl_3_, δ, ppm): 14.42 (СН_3_-20), 17.93 (С-11), 18.96 (C-6), 19.26 (С-2), 22.41 (С-12), 28.85 (СН_3_-19), 33.07 (C-7), 37.87 (C-3)*, 37.89 (C-1)*, 40.04 (C-17)**, 40.28 (C-8)**, 41.67 (С-10), 43.77 (C-4), 55.05 (ОСН_3_), 57.23 (С-9), 57.33 (С-5), 109.77 (С-14), 111.57 (С-4′), 114.87 (С-13), 115.38 (С-2′), 122.81 (С-6′), 128.19 (С-5′), 139.90 (С-15), 156.62 (С-16), 141.16 (С-1′), 158.88 (С-3′), 183.39 (С-18); UV (EtOH) λ_max_, (lgε): 216 (4.08), 274 (3.44), 281 (3.41) nm; IR (KBr, ν, cm^−1^): 3400 (OH), 1693 (C=O); 1454, 779, 694 (Ar); 1263, 1049 (С-О); HR-MS, calcd. for C_27_H_34_O_4_: 422.2452; found, [M]^+^ *m*/*z*: 422.2447.

(5a*R*,6*S*,9a*R*,9b*R*)-6,9a-Dimethyl-3b-(2-methylbenzyl)-3b,4,5,5a,6,7,8,9,9a,9b,10,11-dodecahydrophenanthro [1,2–*b*]furan-6-carboxylic acid (**7i**). Yield 48% (e). White solid. m.p. 223 °С (decomp.); [α]_D_^25^ −11.2 (*c* 0.5, EtOH); ^1^H-NMR (400 MHz, CDCl_3_, δ, ppm): 0.95 (1H, m, H-1*α*), 0.99 (3Н, s, СН_3_-20), 1.05 (1H, dt, *J* = 12.8, 3.2 Hz, H-3*α*), 1.22–1.25 (2Н, m, H-5*α*, H-7), 1.29 (3Н, s, СН_3_-19), 1.43 (1Н, d, *J* = 11.1 Hz, Н-9), 1.50 (1Н, dm, *J* = 14.6 Hz, Н-2), 1.80 (3Н, s, СН_3_), 1.80–2.03 (5H, m, Н-11,6,11,1*β*,2), 2.19 (1Н, dm, *J* = 12.9 Hz, Н-3*β*), 2.14–2.26 (1Н, m, Н-12), 2.35–2.45 (2Н, m, Н-6,7), 2.54–2.61 (1Н, m, Н-12), 2.95 (1Н, d, *J* = 13.0 Hz, Н-17), 3.10 (1Н, d, *J* = 13.0 Hz, Н-17), 6.02 (1Н, s, Н-14), 6.87 (1Н, s, Н-15), 6.97 (1Н, m, Н-6′), 7.01 (1Н, m, Н-5′), 7.07 (2Н, m, Н-3′,4′); ^13^C-NMR (125 MHz, CDCl_3_, δ, ppm): 14.39 (C-20), 17.98 (С-11), 18.74 (СН_3_), 18.93 (C-6), 19.37 (С-2), 22.71 (С-12), 28.76 (C-19), 33.31 (C-7), 35.69 (C-17), 37.80 (C-3)*, 37.91 (C-1)*, 40.04 (C-8), 41.69 (С-10), 43.79 (C-4), 57.39 (С-9), 57.88 (С-5), 109.49 (С-14), 115.04 (С-13), 124.83 (С-4′), 125.79 (С-5′), 129.84 (С-3′), 131.46 (С-6′), 137.35 (С-2′), 137.80 (С-1′), 140.29 (С-15), 156.92 (С-16), 184.22 (С-18); UV (EtOH) λ_max_, (lgε): 210 (4.07), 265 (2.91), 273 (2.86) nm; IR (KBr, ν, cm^−1^): 3440 (OH), 1695 (C=O); 1591, 1462, 752, 733 (Ar); HR-MS, calcd. for C_27_H_34_O_3_: 406.2503; found, [M]^+^ *m*/*z*: 406.2506.

(5a*R*,6*S*,9a*R*,9b*R*)-6,9a-Dimethyl-3b-(3,4,5-trimethoxybenzyl)-3b,4,5,5a,6,7,8,9,9a,9b,10,11-dodecahydrophenanthro [1,2–*b*]furan-6-carboxylic acid (**7j**). Yield: 61% (e), 72% (f). White solid. m.p. 75 °С (decomp.); [α]_D_^25^ +0.4 (*c*=1.1, EtOH); ^1^H-NMR (300 MHz, CDCl_3_, δ, ppm): 0.89–1.01 (1H, m, H-1*α*), 0.96 (3Н, s, СН_3_-20), 1.05 (1H, dt, *J* = 13.2, 3.6 Hz, H-3*α*), 1.21–1.32 (2Н, m, H-5*α*,7), 1.28 (3Н, s, СН_3_-19), 1.42 (1Н, d, *J* = 11.7 Hz, Н-9), 1.49 (1Н, dm, *J* = 15.8 Hz, Н-2), 1.78–1.96 (5H, m, Н-11,6,11,1*β*,2), 2.18 (1Н, dm, *J* = 14.2 Hz, Н-3*β*), 2.15–2.26 (1Н, m, Н-12), 2.33–2.44 (2Н, m, Н-6,7), 2.58 (1Н, dd, *J* = 16.1, 5.9 Hz, Н-12), 2.73 (1Н, d, *J* = 12.8 Hz, Н-17), 2.96 (1Н, d, *J* = 12.8 Hz, Н-17), 3.69 (6Н, s, 2ОСН_3_), 3.79 (3Н, s, ОСН_3_), 5.91 (2Н, s, С-2′,6′), 6.07 (1Н, d, *J* = 1.5 Hz, Н-14), 6.88 (1Н, d, *J* = 1.5 Hz, Н-15); ^13^C-NMR (125 MHz, CDCl_3_, δ, ppm): 14.36 (CH_3_-20), 17.95 (С-11), 18.92 (C-6), 19.22 (С-2), 22.38 (С-12), 28.82 (CH_3_-19), 33.31 (C-7), 37.84 (C-3)*, 37.87 (C-1)*, 39.95 (C-17)**, 40.42 (C-8)**, 41.75 (С-10), 43.80 (C-4), 55.92 (2ОСН_3_), 57.14 (С-9), 57.27 (С-5), 60.86 (ОСН_3_), 104.45 (С-2′,6′), 109.78 (С-14), 114.98 (С-13), 135.32 (С-4′), 137.57 (С-1′), 139.98 (С-15), 152.26 (С-3′,5′), 156.36 (С-16), 184.08 (С-18); UV (EtOH) λ_max_, (lgε): 274 (3.64) nm; IR (KBr, ν, cm^−1^): 3390, 1693 (CO_2_H); 1464, 1419, 754 (Ar); 1007, 1228 (С-О); HR-MS, calcd. for C_29_H_38_O_6_: 482.2663; found, [M]^+^ *m*/*z*: 482.2665.

(1*S*,4a*R*,5*S*,8a*R*)-1,4a-Dimethyl-6-methylene-5-(2-(2-(3,4,5-trimethoxyphenyl)furan-3-yl)ethyl)decahydronaphthalene-1-carboxylic acid (**8j**). Yield 8% (e). White solid. m.p. 88 °С (decomp.); [α]_D_^25^ +21.8 (*c* 0.6, EtOH); ^1^H-NMR (300 MHz, CDCl_3_, δ, ppm): 0.57 (3Н, s, СН_3_-20), 0.95 (m, 1H, H-1*α*), 0.99 (m, 1H, H-3*α*), 1.20 (3Н, s, СН_3_-19), 1.26 (1Н, m, H-5*α*), 1.45 (1Н, dm, *J* = 13.8 Hz, Н-2), 1.64 (2Н, m, Н-11,9), 1.72–1.93 (6H, m, Н-6,11,6,7,1*β*,2), 2.11 (1Н, dm, *J* = 13.3 Hz, Н-3*β*), 2.38 (1Н, dm, *J* = 6.7 Hz, Н-7), 2.55 (1Н, m, Н-12), 2.72 (1Н, m, Н-12), 3.88 (3Н, s, ОСН_3_), 3.87 (6Н, s, 2ОСН_3_), 4.62 (1Н, s, Н-17), 4.87 (1Н, s, Н-17), 6.35 (1Н, d, *J* = 1.6 Hz, Н-14), 6.77 (2Н, s, Н-2′,6′), 7.36 (d, 1Н, *J* = 1.6 Hz, Н-15); ^13^C-NMR (150 MHz, CDCl_3_, δ, ppm): 12.76 (CH_3_-20), 19.84 (С-2), 24.26 (С-12), 24.36 (С-11), 25.98 (C-6), 28.94 (CH_3_-19), 37.83 (C-3), 38.66 (C-7), 39.06 (C-1), 40.44 (C-4), 44.15 (С-10), 55.05 (С-9), 56.16 (С-5), 56.16 (2ОСН_3_), 60.93 (ОСН_3_), 103.29 (С-2′_,_6′), 106.59 (С-17), 113.39 (С-14), 121.18 (С-13), 127.47 (С-1′), 137.32 (С-4′), 140.81 (С-15), 147.99 (С-8), 148.53 (С-16), 153.27 (С-3′_,_5′), 183.75 (С-18); UV (EtOH) λ_max_, (lgε): 280 (4.02) nm; IR (KBr, ν, cm^−1^): 3322 (OH), 1693 (C=O), 1594, 1416, 754 (Ar), 1238 (С-О); HR-MS, calcd. for C_29_H_38_O_6_: 482.2663; found, [M]^+^ *m*/*z*: 482.2668.

### 3.3. Synthesis of 15-Substituted 16-Bromolambertianic Acid Derivatives Substituted (***9***–***12***)

(1*S*,4a*R,5S*,8a*R*)-5-(2-(2-Bromo-5-formylfuran-3-yl)ethyl)-1,4a-dimethyl-6-methylenedecahydronaphthalene-1-carboxylic acid [16-Bromo-15-formyllambertianic acid] (**9**). To a cold (0 °C) stirred solution of 16-bromolambertianic acid (**5**) (0.50 g, 1.58 mmol) in dimethylformamide (10 mL), phosphoryl chloride (2.06 mL, 22.78 mmol) was added dropwise at 0 °C; then, the reaction mixture was left to stand for 48 h at rt. The mixture was then poured into ice water (40 mL), and saturated aqueous solution of sodium acetate (20 mL) was added. The organic phase was separated, and the aqueous phase was extracted with chloroform (3 × 30 mL). The combined organic solution was washed with water (3 × 15 mL), dried over MgSO_4_, filtered and evaporated in vacuo. The residue was subjected to column chromatography on silica gel (eluent petroleum ether-Et_2_O, 2:1) to isolate compound **9** (0.48 g, 89% yield) as a brownish solid. m.p. 87.7 °С (decomp.). ^1^H-NMR (300 MHz, CDCl_3_, δ, ppm): 0.53 (3Н, s, СН_3_-20), 0.92 (1H, m, H-1*α*), 0.96 (1H, m, H-3*α*), 1.16 (3Н, s, СН_3_-19), 1.24 (1Н, d, *J* = 10.1 Hz, H-5*α*), 1.43 (1Н, dm, *J* = 12.6 Hz, Н-2), 1.52 (2Н, m, Н-11,9), 1.66-1.85 (5H, m, Н-11,6,1*β*,2,7), 1.88–1.93 (1Н, m, Н-6), 2.07 (1Н, dm, *J* = 13.0 Hz, Н-3*β*), 2.22 (1Н, m, Н-12), 2.36 (1Н, dm, *J* = 6.5 Hz, Н-7), 2.51 (1Н, m, Н-12), 4.52, 4.85 (2Н, both s, Н-17), 7.10 (1Н, s, Н-14), 9.42 (1Н, s, CHO), 11.78 (1Н, br.s, OH); ^13^C-NMR (125 MHz, CDCl_3_, δ, ppm): 12.57 (СН_3_-20), 19.64 (С-2), 22.84 (С-12), 23.30 (С-11), 25.81 (C-6), 28.77 (СН_3_-19), 37.55 (C-3), 38.42 (C-7), 38.83 (C-1), 40.22 (C-4), 43.97 (С-10), 54.86 (С-9), 55.96 (С-5), 106.47 (С-17), 123.16 (С-14), 123.70 (С-13), 140.86 (С-16), 147.13 (С-8), 150.78 (С-15), 176.32 (CНО), 184.19 (С-18); UV (EtOH) λ_max_, (lgε): 224 (3.64), 292 (3.04) nm; IR (film, ν, cm^−1^): 3340 (OH), 1689 (C=O), 1468, 1383, 1035 (furan), 891 (C=CН_2_), 612 (Br); HR-MS, calcd. for C_21_H_26_O_4_ [M-Br]^+^: 342.1826; found, [M-Br]^+^ *m*/*z*: 342.1822.

(1*S*,4a*R*,5*S*,8a*R*)-5-(2-(2-Вromo-5-cyanofuran-3-yl)ethyl)-1,4a-dimethyl-6-methylenedecahydronaphthalene-1-carboxylic acid [16-Вromo-15-cyanolambertianic acid] (**10**). A water solution of ammonia (5 mL, 70 (mmol) and iodine (0.60 g, 2.36 mmol) were added to a vigorously stirred solution of 16-bromo-15-formyllambertianic acid **9** (0.50 g, 1.18 mmol) in THF (10 mL). The mixture was stirred at room temperature for 24 h; then, it was treated with diluted H_2_SO_4_ (0.1 mL in 1 mL of H_2_O) to pH 5 and extracted with CHCl_3_ (3 × 50 mL). The combined organic solution was washed with water (3 × 15 mL), dried over MgSO_4_, filtered and evaporated in vacuo. The residue was purified by column chromatography on silica gel (eluent petroleum ether-Et_2_O, 2:1) to afford compound **10** (0.46 g, 93% yield) as a white solid, m.p. 55.1 °С (decomp.); [α]_D_^25^ +44.3 (*c* 0.4, EtOH). ^1^H-NMR (300 MHz, CDCl_3_, δ, ppm): 0.55 (3Н, s, СН_3_-20), 0.95 (1H, m, H-1*α*), 0.99 (1H, m, H-3*α*), 1.19 (3Н, s, СН_3_-19), 1.27 (1Н, d, *J* = 10.5 Hz, H-5*α*), 1.44–1.59 (3Н, m, Н-11,9,2), 1.66–1.94 (6H, m, Н-11,6,1*β*,2,7,6), 2.10 (1Н, dm, *J* = 13.2 Hz, Н-3*β*), 2.22 (1Н, m, Н-12), 2.38 (1Н, dm, *J* = 6.7 Hz, Н-7), 2.50 (1Н, m, Н-12), 4.52, 4.87 (2Н, both s, Н-17), 6.97 (1Н, s, Н-14), 12.18 (1Н, br.s, OH); ^13^C-NMR (75 MHz, CDCl_3_, δ, ppm): 12.55 (СН_3_-20), 19.64 (С-2), 22.72 (С-12), 23.18 (С-11), 25.80 (C-6), 28.75 (СН_3_-19), 37.52 (C-3), 38.40 (C-7), 38.83 (C-1), 40.22 (C-10), 43.99 (С-4), 54.83 (С-9), 55.94 (С-5), 106.49 (С-17), 110.61 (C≡N), 122.27 (С-13), 124.45 (С-15), 124.58 (С-14), 138.18 (С-16), 147.04 (С-8), 184.45 (С-18); UV (EtOH) λ_max_, (lgε): 258 (4.11) nm; IR (KBr, ν, cm^−1^): 3400 (OH), 1693 (C=O), 2229 (C≡N), 1468, 1377, 1032 (furan); 1645, 1410, 891 (C=CН_2_), 610 (Br); HR-MS, calcd. for [*M-Br*]^+^ C_21_H_26_O_3_N: 340.1995; found, [*M-Br*]^+^
*m*/*z*: 340.1990.

(1*S*,4a*R*,5*S*,8a*R*)-5-(2-(2-Bromo-5-(hydroxymethyl)furan-3-yl)ethyl)-1,4a-dimethyl-6-methylenedecahydronaphthalene-1-carboxylic acid [16-Bromo-15-(hydroxymethyl) lambertianic acid] (**11**). NaBH_4_ (0.45 g, 1.18 mmol) was added portion wise to a solution of 16-bromo-15-formyllambertianic acid **9** (0.50 g, 1.18 mmol) in MeOH (10 mL) under stirring at 20 °C. The mixture was stirred at room temperature for 24 h; then, it was treated with diluted H_2_SO_4_ (0.1 mL in 1 mL of H_2_O) to pH 5 under stirring at room temperature. The reaction product was extracted with CHCl_3_ (5 × 50 mL). The organic solution was washed with water (3 × 15 mL), dried over MgSO_4_, filtered and evaporated in vacuo. The residue was purified by column chromatography (eluent CHCl_3_-MeOH, 50:1) to afford compound **11** (0.36 g, 71% yield) as a white solid. m.p. 74.8 °С (decomp.); [α]_D_^25^ +40.7 (*c* 0.6, EtOH). ^1^H-NMR (300 MHz, CDCl_3_, δ, ppm): 0.53 (3Н, s, СН_3_-20), 0.98 (2H, m, H-1*α*,3*α*), 1.19 (3Н, s, СН_3_-19), 1.27 (1Н, d, *J* = 10.7 Hz, H-5*α*), 1.44–1.54 (3Н, m, Н-11,9,2), 1.58–1.96 (6H, m, Н-11,6,1*β*,2,7,6), 2.07–2.17 (1Н, m, Н-12), 2.10 (1Н, dm, *J* = 13.7 Hz, Н-3*β*), 2.40 (2Н, m, Н-7, Н-12), 4.46 (2Н, s, СН_2_), 4.55, 4.87 (2Н, both s, Н-17), 6.16 (1Н, s, Н-14); ^13^C-NMR (125 MHz, CDCl_3_, δ, ppm): 12.70 (СН_3_-20), 19.77 (С-2), 23.17 (С-12), 23.66 (С-11), 25.95 (C-6), 28.91 (СН_3_-19), 37.74 (C-3), 38.57 (C-7), 38.92 (C-1), 40.31 (C-10), 44.10 (С-4), 55.10 (С-9), 56.10 (С-5), 57.24 (СН_2_), 106.49 (С-17), 111.09 (С-14), 120.44 (С-13), 132.44 (С-16), 147.50 (С-8), 152.31 (С-15), 184.12 (С-18); UV (EtOH) λ_max_, (lgε): 228 (3.80), 289 (2.96) nm; IR (KBr, ν, cm^−1^): 3412 (OH), 1693 (C=O), 1468, 1385, 1030 (furan); 1645, 1406, 891 (C=CН_2_), 615 (Br); HR-MS, calcd. for C_21_H_29_O_4_Br: 424.1244; found, [*Mr*]^+^
*m*/*z*: 424.1241.

(1*S*,4a*R*,5*S*,8a*R*)-5-(2-(2-Bromo-5-((methylamino)methyl)furan-3-yl)ethyl)-1,4a-dimethyl-6-methylenedecahydronaphthalene-1-carboxylic acid [16-Bromo-15-((methylamino) methyl)lambertianic acid] (**12**). A solution of metylamine hydrochloride (0.19 g, 2.84 mmol) in CH_2_Cl_2_ (10 mL) was treated with Et_3_N (0.78 mL, 5.67 mmol) at room temperature. After stirring for 30 min, 16-bromo-15-formyllambertianic acid **9** (0.30 g, 0.71 mmol) and MgSO_4_ (2.00 g) were added, and stirring was continued for another 48 h at room temperature. The reaction mixture was filtrated off, and the solvent was removed under reduced pressure. The residue was dissolved in methanol (30 mL), and sodium borohydride (0.08 g, 2.13 mmol) was added portion wise. After 10 h stirring, water (30 mL) was added; then, the reaction product was extracted with ethyl acetate (3 × 40 mL). The combined organic solution was washed with water (3 × 40 mL), dried over MgSO_4_, and filtered. The solvent was distilled off, and the residue was subject to the column chromatography on silica gel (eluent: CHCl_3_-MeOH, 20:1) to afford compound **12** (0.22 g, 70%yield) as a white solid. m.p. 132.1 °С (decomp.); [α]_D_^25^ +40.6 (*c* 0.6 in EtOH). ^1^H-NMR (500 MHz, CDCl_3_, δ, ppm): 0.54 (3Н, s, СН_3_-20), 0.95 (2H, m, H-1*α*,3*α*), 1.13 (3Н, s, СН_3_-19), 1.21 (1Н, m, H-5*α*), 1.42 (1Н, m, H-2), 1.55–1.65 (2Н, m, Н-11,9), 1.70–1.88 (5H, m, Н-11,6,1*β*,2,7), 1.91–2.07 (m, 2Н, Н-12,6), 2.08 (1Н, dm, *J* = 12.6 Hz, Н-3*β*), 2.37–2.46 (2Н, m, Н-7,12), 2.42 (3Н, s, СН_3_), 3.75 (2Н, s, СН_2_), 4.52, 4.84 (2Н, both s, Н-17), 6.22 (1Н, s, Н-14); ^13^C-NMR (75 MHz, CDCl_3_, δ, ppm): 12.81 (СН_3_-20), 19.89 (С-2), 23.32 (С-12), 23.71 (С-11), 26.05 (C-6), 29.02 (СН_3_-19), 37.98 (C-3), 38.67 (C-7), 39.10 (C-1), 40.36 (C-10), 40.74 (СН_3_), 44.13 (С-4), 51.72 (СН_2_), 55.30 (С-9), 56.23 (С-5), 106.50 (С-17), 112.89 (С-14), 120.49 (С-13), 132.06 (С-16), 147.61 (С-8), 149.20 (С-15), 183.23 (С-18); UV (EtOH) λ_max_, (lgε): 228 (3.91) nm; IR (KBr, ν, cm^−1^): 3437 (OH, NH), 1701 (C=O), 1549, 1169 (NHMe), 1468, 1030 (furan), 1643, 887 (C=CН_2_), 608 (Br); HR-MS, calcd. for C_22_H_32_O_3_NBr: 437.1560; found, [*Mr*]^+^
*m*/*z*: 437.1558.

### 3.4. Synthesis of 15-Substituted 16-Aryllambertianic Acid Derivatives (***13a***–***c***,***14***)

(1*S*,4a*R*,5*S*,8a*R*)-5-(2-(5-Formyl-2-phenylfuran-3-yl)ethyl)-1,4a-dimethyl-6-methylenedecahydronaphthalene-1-carboxylic acid (**13a**). A mixture of 16-bromo-15-formyllambertianic acid **9** (0.30 g, 0.71 mmol), phenylboronic acid **6a** (0.10 g, 0.85 mmol), Pd(OAc)_2_ (0.01 g, 0.04 mmol), (*R*)-BINAP (0.05 g, 0.08 mmol) and K_2_CO_3_ (0.38 g, 2.75 mmol) in DMF (3 mL) and H_2_O (1.5 mL) was stirred at 80–85 °C (bath) for 24 h under a stream of argon. After cooling, the mixture was treated with diluted H_2_SO_4_ (0.1 mL in 1 mL of H_2_O) to pH 5 and extracted with CHCl_3_ (3 × 50 mL). The combined organic solution was washed with water (3 × 15 mL), dried over MgSO_4_ and evaporated in vacuo. The residue was purified by column chromatography on silica gel (eluent petroleum ether-Et_2_O, 2:1) to afford compound **13a** (0.21 g, 69%) as a white solid. m.p. 73.3 °С (decomp.); [α]_D_^25^ +28.7 (*c* 0.5, EtOH). ^1^H-NMR (500 MHz, CDCl_3_, δ, ppm): 0.58 (3Н, s, СН_3_-20), 0.95 (1H, m, H-1*α*), 1.00 (1H, dt, *J* = 13.6, 3.8 Hz, H-3*α*), 1.20 (3Н, s, СН_3_-19), 1.27 (1Н, dd, *J* = 11.8, 3.1 Hz, H-5*α*), 1.46 (1Н, m, Н-2), 1.56–1.64 (2Н, m, Н-11, 9), 1.67–1.87 (5H, m, Н-11,6,1*β*,2,7), 1.96 (1Н, m, Н-6), 2.11 (1Н, dm, *J* = 13.4 Hz, Н-3*β*), 2.40 (1Н, dm, *J* = 8.2 Hz, Н-7), 2.56 (1Н, m, Н-12), 2.82 (1Н, m, Н-12), 4.57, 4.89 (2Н, both s, Н-17), 7.22 (1Н, s, Н-14), 7.37 (1Н, t, *J* = 8.2 Hz, Н-4′), 7.42 (2Н, t, *J* = 8.2 Hz, Н-3′,5′), 7.70 (d, 2Н, *J* = 8.2 Hz, Н-2′,6′), 9.60 (s, 1Н, CHO), 11.78 (brs, 1Н, OH); ^13^C-NMR (150 MHz, CDCl_3_, δ, ppm): 12.78 (СН_3_-20), 19.79 (С-2), 23.95 (С-12), 24.38 (С-11), 25.96 (C-6), 28.93 (СН_3_-19), 37.75 (C-3), 38.56 (C-7), 39.01 (C-1), 40.48 (C-10), 44.14 (С-4), 55.15 (С-9), 56.16 (С-5), 106.68 (С-17), 122.82 (С-14), 124.82 (С-13), 126.80 (С-2′,6′), 128.72 (С-3′,5′), 129.00 (С-4′), 129.92 (С-1′), 147.64 (С-8), 150.72 (С-16), 154.69 (С-15), 177.53 (СНО), 183.83 (С-18); UV (EtOH) λ_max_, (lgε): 228 (4.08), 324 (4.07) nm; IR (KBr, ν, cm^−1^): 3440 (OH), 1689, 1680 (C=O, СНО), 1483, 1446, 756, 694 (Ph), 1470, 1385, 1030 (furan), 1412, 891 (C=CН_2_); HR-MS, calcd. for C_27_H_32_O_4_: 420.2290; found, [M]^+^ *m*/*z*: 420.2295.

(1*S*,4a*R*,5*S*,8a*R*)-5-(2-(2-(4-Cyanophenyl)-5-formylfuran-3-yl)ethyl)-1,4a-dimethyl-6-methylenedecahydronaphthalene-1-carboxylic acid (**13b**). The reaction of 16-bromo- 15-formyllambertianic acid **9** (0.30 g, 0.71 mmol) with (4-cyanophenyl)boronic acid **6f** (0.13 g, 0.85 mmol) under the conditions of the preparation of compound **13a** gave compound **13b** (0.18 g, 58%) as a white solid. mp 97.5 °С (decomp.); [α]_D_^25^ +12.4 (*c* 0.7, EtOH); ^1^H-NMR (300 MHz, CDCl_3_, δ, ppm): 0.58 (3Н, s, СН_3_-20), 0.94 (1H, m, H-1*α*), 0.99 (1H, dt, *J* = 13.4, 3.8 Hz, H-3*α*), 1.19 (3Н, s, СН_3_-19), 1.26 (1Н, m, H-5*α*), 1.46 (1Н, m, Н-2), 1.58–1.62 (2Н, m, Н-11,9), 1.70–1.88 (5H, m, Н-11,6,1*β*,2,7), 1.94 (1Н, m, Н-6), 2.11 (1Н, dm, *J* = 13.1 Hz, Н-3*β*), 2.40 (1Н, dm, *J* = 8.0 Hz, Н-7), 2.58 (1Н, m, Н-12), 2.84 (1Н, m, Н-12), 4.55, 4.90 (2Н, both s, Н-17), 7.23 (1Н, s, Н-14), 7.69 (2Н, d, *J* = 8.1 Hz, Н-3′,5′), 7.81 (2Н, d, *J* = 8.1 Hz, Н-2′,6′), 9.64 (1Н, s, CHO), 11.76 (1Н, br.s, OH); ^13^C-NMR (100 MHz, CDCl_3_, δ, ppm): 12.72 (СН_3_-20), 19.72 (С-2), 23.71 (С-12), 24.41 (С-11), 25.88 (C-6), 28.86 (СН_3_-19), 37.62 (C-3), 38.49 (C-7), 38.98 (C-1), 40.46 (C-10), 44.07 (С-4), 55.09 (С-9), 56.05 (С-5), 106.68 (С-17), 111.98 (С-4′), 118.34 (C≡N), 126.81 (С-2′,6′), 124.01 (С-14), 127.24 (С-13), 132.45 (С-3′,5′), 133.89 (С-1′), 147.55 (С-8), 151.29 (С-16), 151.80 (С-15), 177.72 (СНО), 183.96 (С-18); UV (EtOH) λ_max_, (lgε): 237 (4.21), 329 (4.38) nm; IR (KBr, ν, cm^−1^): 3400 (OH), 2227 (C≡N), 1682 (CO_2_H, СНО), 1487, 1448, 845 (Ph), 1468, 1385, 1035 (furan), 1418, 891 (C=CН_2_); HR-MS, calcd. for C_28_H_31_O_4_N: 445.2248; found, [M]^+^ *m*/*z*: 445.2242.

(1*S*,4a*R*,5*S*,8a*R*)-5-(2-(5-Formyl-2-(3,4,5-trimethoxyphenyl)furan-3-yl)ethyl)-1,4a-di-methyl-6-methylenedecahydronaphthalene-1-carboxylic acid (**13c**). The reaction of 16- bromo-15-formyllambertianic acid **9** (0.30 g, 0.71 mmol) with (3,4,5-trimethoxyphenyl)boronic acid **6j** (0.18 g, 0.85 mmol) under the conditions of the preparation of compound **13a** afforded compound **13c** (0.18 g, 49%) as a yellowish oil. [α]_D_^25^ +14 (*c* 0.9, EtOH); ^1^H-NMR (400 MHz, CDCl_3_, δ, ppm): 0.57 (3Н, s, СН_3_-20), 0.92 (1H, m, H-1*α*), 0.98 (1H, dt, *J* = 13.4, 3.5 Hz, H-3*α*), 1.19 (3Н, s, СН_3_-19), 1.26 (1Н, m, H-5*α*), 1.45 (1Н, m, Н-2), 1.58-1.61 (2Н, m, Н-11,9), 1.64–1.89 (5H, m, Н-11,6,1*β*,2,7), 1.93 (1Н, m, Н-6), 2.10 (1Н, dm, *J* = 13.2 Hz, Н-3*β*), 2.37 (1Н, dm, *J* = 8.9 Hz, Н-7), 2.58 (1Н, m, Н-12), 2.79 (1Н, m, Н-12), 3.86 (3H, s, OCH_3_), 3.87 (s, 6H, 2OCH_3_), 4.57, 4.87 (2Н, both s, Н-17), 6.88 (2Н, s, Н-2′,6′), 7.20 (1Н, s, Н-14), 9.58 (1Н, s, CHO); ^13^C-NMR (125 MHz, CDCl_3_, δ, ppm): 12.70 (СН_3_-20), 19.72 (С-2), 23.95 (С-12), 24.21 (С-11), 25.88 (C-6), 28.88 (СН_3_-19), 37.65 (C-3), 38.53 (C-7), 39.02 (C-1), 40.40 (C-10), 44.05 (С-4), 54.93 (С-9), 56.03 (С-5), 56.23 (2OCH_3_), 60.94 (OCH_3_), 104.24 (С-2′,6′), 106.60 (С-17), 122.94 (С-14), 124.37 (С-13), 125.34 (С-1′), 138.87 (С-4′), 147.66 (С-8), 150.44 (С-16), 153.32 (С-3′,5′), 154.70 (С-15), 177.34 (СНО), 183.89 (С-18); UV (EtOH) λ_max_, (lgε): 214 (4.38), 237 (3.95), 256 (3.81), 339 (4.15) nm; IR (film, ν, cm^−1^): 3410 (OH), 1678 (C=O, СНО), 1493, 756 (Ph), 1385, 1032 (furan), 1417, 887 (C=CН_2_), 1241, 1003 (С-О); HR-MS, calcd. for C_30_H_38_O_7_: 510.2612; found, [M]^+^ *m*/*z*: 510.2609.

(1*S*,4a*R*,5*S*,8a*R*)-5-(2-(5-cyano-2-phenylfuran-3-yl)ethyl)-1,4a-dimethyl-6-methylenedecahydronaphthalene-1-carboxylic acid (**14**). The reaction of 16-bromo-15-cyanolambertianic acid **10** (0.30 g, 0.71 mmol) with phenylboronic acid **6a** (0.10 g, 0.85 mmol) under the conditions of the preparation of compound **13a** afforded compound **14** (0.15 g, 52%) as a white solid. m.p. 95.3 °С (decomp.); [α]_D_^25^ +33.2 (*c* 0.7, EtOH); ^1^H-NMR (400 MHz, CDCl_3_, δ, ppm): 0.57 (3Н, s, СН_3_-20), 0.97 (2H, m, H-1*α*,3*α*), 1.14 (3Н, s, СН_3_-19), 1.22 (1Н, m, H-5*α*), 1.43 (1Н, m, Н-2), 1.54–1.67 (2Н, m, Н-11,9), 1.72–1.89 (5H, m, Н-11,6,1*β*,2,7), 1.93 (1Н, m, Н-6), 2.07 (1Н, dm, *J* = 13.3 Hz, Н-3*β*), 2.36 (1Н, m, Н-7), 2.50 (1Н, m, Н-12), 2.75 (1Н, m, Н-12), 4.53, 4.84 (2Н, both s, Н-17), 7.10 (1Н, s, Н-14), 7.30 (1Н, t, *J* = 7.6 Hz, Н-4′), 7.38 (2Н, m, Н-3′,5′), 7.60 (2Н, d, *J* = 7.6 Hz, Н-2′, Н-6′); ^13^C-NMR (125 MHz, CDCl_3_, δ, ppm): 12.33 (СН_3_-20), 19.58 (С-2), 23.74 (С-12), 24.05 (С-11), 25.84 (C-6), 28.63 (СН_3_-19), 37.76 (C-3), 38.31 (C-7), 38.80 (C-1), 40.08 (C-10), 43.66 (С-4), 55.99 (С-9), 55.78 (С-5), 106.07 (С-17), 118.57 (С-14), 123.98 (C≡N), 126.03 (С-2′,6′), 127.94 (С-4′), 128.32 (С-3′,5′), 130.08 (С-13), 144.43 (С-1′), 147.62 (С-8), 151.00 (С-16), 161.17 (С-15), 180.16 (С-18); UV (EtOH) λ_max_, (lgε): 216 (4.13), 262 (3.79), 297 (3.94) nm; IR (KBr, ν, cm^−1^): 3479, 1693 (CO_2_H); 2224 (CN); 1446, 758, 696 (Ph), 1468, 1030 (furan); 891 (C=CН_2_); HR-MS, calcd. for C_27_H_31_O_3_N: 417.2299; found, [M]^+^ *m*/*z*: 417.2286.

## 4. Conclusions

In summary, we have described an efficient protocol for the Heck cyclization–Suzuki coupling cascade reaction toward the synthesis of 17-aryl derivatives of marginatafuran-type isospongian diterpenoids. The latter are formed as an individual stereoisomer, with an axial arrangement of the aryl substituent, using both chiral and achiral phosphine ligands. (*R*)-BINAP has been shown to be the most efficient chiral phosphine ligand. The choice of the base (K_2_CO_3_, K_3_PO_4_, Cs_2_CO_3_) and solvent (DMF-H_2_O, CH_3_CN-H_2_O) is crucial to lead the direction of the reaction. Electron-rich and electron-deficient arylboronic acids can be used in this domino reaction, although coupling with arylboronic acid containing electron-withdrawing groups provides higher yields of tetracyclic compound. The study of the influence of the nature of the substituent in the furan ring of 16-bromolambertianate showed that derivatives with an electron-donating substituent, under the conditions found, are not active in cross-coupling reactions, and derivatives with an electron-withdrawing substituent give exclusively the furan ring arylation products.

Overall, this domino process allows the synthesis of interesting 17-arylsubstituted marginatafuran-type isospongians, which is a common structural motif among a rare group of biologically interesting diterpenoids.

## Data Availability

Data regarding synthesis, isolation and characterization are available upon request from Y.V.K.

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
