# Peer review of "An Approach toward 17-Arylsubstituted Marginatafuran-Type Isospongian Diterpenoids via a Palladium-Catalyzed Heck–Suzuki Cascade Reaction of 16-Bromolambertianic Acid"

_molecules, 2022, doi:10.3390/molecules27092643_

Round 1

Reviewer 1 Report

The manuscript (molecules-1682495 ) entitled "An approach toward 17-arylsubstituted marginatafuran-type isospongian diterpenoids via a palladium-catalyzed Heck–Suzuki cascade reaction of 16-bromolambertianic acid" by Y.V. Kharitonov and E.E. Shults submitted to the journal Molecules for consideration describes the synthesis and analytical characterization of a number of chemical compounds (biologically relevant). The syntheses are meticulously and thoroughly described. All necessary analytical methods are employed (combustion analysis, NMR, IR, and HRMS). Copies of spectra are included as evidence of purity and correct chemical structure. The introduction poses the topic of the article in a very good way. The language and editing of the paper are of a very high standard. Due to the high scientific level, I propose to publish this manuscript after taking into account the following few technical issues. I have no objections to the scientific side, I see no factual errors. 

Technical problems to be improved:
Figure 1; Table 1; Scheme 2, 5: Substituent designations. Correct the superscript and subscript. 
Scheme 3: Some words are unreadable by overlapping text.
Scheme 4: The "70%" in the way "d" overlaps the arrow.
Supporting Materials:
All structural formulas (including atomic numbering) are double overlaid. 
P. 12: Poor resolution of spectrum image.
PP 45 and 46: On previous pages, the chemical name, formula, and spectrum of a compound were shown on single page. In this case, the chemical name is on one page, while the formula and spectrum are on the other. Please put it on single page.

Author Response

 Authors were very grateful for the valuable remarks from Referee 1. We have worked through all comments carefully and made the corresponding changes to the manuscript for better presentation of our results.

  1. Figure 1; Table 1; Scheme 2, 5: Substituent designations. Correct the superscript and subscript.

 Figure, Table and scheme were corrected.

Scheme 3: Some words are unreadable by overlapping text.

The scheme was corrected.
Scheme 4: The "70%" in the way "d" overlaps the arrow.
The scheme was corrected.
Supporting Materials:
All structural formulas (including atomic numbering) are double overlaid. 
P. 12: Poor resolution of spectrum image.
PP 45 and 46: On previous pages, the chemical name, formula, and spectrum of a compound were shown on single page. In this case, the chemical name is on one page, while the formula and spectrum are on the other. Please put it on single page.

The Supp. materials are corrected.

We made important corrections and additions to the manuscript, which were necessary for the better presentation of our scientific material. Thank you very much for all the comment.   

Yours sincerely,
 Elvira Shults

Reviewer 2 Report

The manuscript submitted by E. E. Shults and co-workers reports a new route for the synthesis of lambertianic acid derivatives. This approach makes use of a Pd-catalyzed cascade process in which a domino Heck and Suzuki coupling take place to deliver the desired compound. As far as I am aware, these cascades have been deeply investigated in the literature, however they have not yet been applied to the synthesis of this type of natural product derivatives. Therefore, they may constitute an interesting piece of work for those working on the field of natural product derivatization. In my opinion the work would be acceptable for its publication in Molecules after major revision, dealing with the following points:

  • The English grammar needs a deep revision and correction in order to make the text grammatically correct and improve its understanding.
  • Some of the labels and captions in the Figure 1, Scheme 3, Scheme 4, Scheme 5, cannot be seen clearly. For example, in Fig. 1 the text of labels 1,2,3 are mixed in several lines. Please correct it for clarity.
  • In the lines 158, 164, the authors mention the term “double cross-coupling”. In my opinion this is not correct, since the first Heck coupling reaction occurs intramolecularly, only the later Suzuki coupling is intermolecular. I would suggest removing the term “double cross-coupling”.
  • In the SI file, the images of the molecules pasted in their corresponding spectra cannot be seen adequately, they seem to be duplicated.
  • The 1H and 13C-NMR spectra contain significant peaks that do not belong to the synthesized products. The authors must comment in the text if these observed peaks belong to minor diastereoisomers present in the natural product, or impurities arising from the isolation of the starting lambertianic acid.
  • Regarding the references section, the pioneering work on Pd-catalyzed cascade Heck-Suzuki reactions by R. Grigg (Tetrahedron 1997, 53, 11803) should be included.

Author Response

Authors were very grateful for the valuable remarks from Referee 2. We have worked through all comments carefully and made the corresponding changes to the manuscript. Please find below all our answers in the text in a point by point manner.

The English grammar needs a deep revision and correction in order to make the text grammatically correct and improve its understanding

According to the Reviewer, we revised the text.

Some of the labels and captions in the Figure 1, Scheme 3, Scheme 4, Scheme 5, cannot be seen clearly. For example, in Fig. 1 the text of labels 1,2,3 are mixed in several lines. Please correct it for clarity.

Thank you very much, al label and captions in the Figures and Schemes were corrected. 

In the lines 158, 164, the authors mention the term “double cross-coupling”. In my opinion this is not correct, since the first Heck coupling reaction occurs intramolecularly, only the later Suzuki coupling is intermolecular. I would suggest removing the term “double cross-coupling”.

Thank you very much for this comments. We have removed the term “double cross-coupling” (p. 5).

In the SI file, the images of the molecules pasted in their corresponding spectra cannot be seen adequately, they seem to be duplicated.

 The images of the molecules has been replaced.

The 1H and 13C-NMR spectra contain significant peaks that do not belong to the synthesized products. The authors must comment in the text if these observed peaks belong to minor diastereoisomers present in the natural product, or impurities arising from the isolation of the starting lambertianic acid.

The main impurities in the spectra are arising from the isolation oa the starting lambertianic acid and belong to the additional natural compounds. Additional purification lead only to a decrease in the yield of the products. Impurities remaining in the spectra after column chromatography. This fact was also comment in the text (p.9 in general information part).

Regarding the references section, the pioneering work on Pd-catalyzed cascade Heck-Suzuki reactions by R. Grigg (Tetrahedron 1997, 53, 11803) should be included.

Thank you very much. The marked Reference by the excellent work from R. Grigg was included in the text. Reference [21].

Thank you very much for all your comments.    We made important corrections and additions to the manuscript, which were necessary for the better presentation of our scientific material.

Yours sincerely,
 Elvira Shults

Reviewer 3 Report

This paper describes a palladium-catalyzed Heck–Suzuki cascade reaction of 16-bromolambertianic acid. This coupling reaction using 2-bromofuran substrate with arylboronic acids provides direct access to the corresponding tetracyclic skeleton of 12 marginatafuran-type isospongian diterpenoids in good to excellent yields. The investigation was carefully achieved, and the paper is well written. I would recommend publication of this manuscript in molecules.

The following points should be considered before the publication.

  • Page 6, line 189: The words “Oxidative addition of palladium(0) to terpenoid bromofuran 5” should be corrected to “Oxidative addition of terpenoid bromofuran 5 to palladium(0)”.
  • Figure 1 and Scheme 1-4: text garbling.

Author Response

The following points should be considered before the publication.

  • Page 6, line 189: The words “Oxidative addition of palladium(0) to terpenoid bromofuran 5” should be corrected to “Oxidative addition of terpenoid bromofuran 5 to palladium(0)”.

  • Figure 1 and Scheme 1-4: text garbling.

Authors were very grateful for the valuable remarks from Referee 3. We have worked through the comments carefully and made the corresponding changes to the manuscript for better presentation of our results.

Page 6, line 189: The words “Oxidative addition of palladium(0) to terpenoid bromofuran 5” should be corrected to “Oxidative addition of terpenoid bromofuran 5 to palladium(0)”.

Thank you very much. This is an important correction.

Figure 1 and Scheme 1-4: text garbling.

 Figure and scheme 1-4 were corrected.

Thank you very much for your comments.   

Yours sincerely,
 Elvira Shults

Round 2

Reviewer 2 Report

The authors have made some good improvements and corrections in the current version of the manuscript. However, I still have found several grammatical errors:

line 43: the text says "was cyclize", should be "was cyclized"

line 116: "the sterically volume" is not correct, could be replaced by "the steric hindrance"

line 124: the text says "we shown", should be "we have shown"

line 165: the text says " in this experiments", should be  "in these experiments"

line 259: the text says "can be explain" should be "can be explained"

Author Response

The authors have made some good improvements and corrections in the current version of the manuscript. However, I still have found several grammatical errors:

line 43: the text says "was cyclize", should be "was cyclized"

line 116: "the sterically volume" is not correct, could be replaced by "the steric hindrance"

line 124: the text says "we shown", should be "we have shown"

line 165: the text says " in this experiments", should be  "in these experiments"

line 259: the text says "can be explain" should be "can be explained"

Authors were very grateful for the additional valuable remarks from Referee 2.Thank you very much for your comments.  All grammatical errors were corrected in the text. 

Yours sincerely,
 Elvira Shults